# Principled Model Routing
# for Unknown Mixtures of Source Domains

**Christoph Dann**
Google Research

**Yishay Mansour**
Google Research
Tel Aviv University

**Teodor V. Marinov**
Google Research

**Mehryar Mohri**
Google Research
Courant Institute

## Abstract

The rapid proliferation of domain-specialized machine learning models presents a challenge: while individual models excel in specific domains, their performance varies significantly across diverse applications. This makes selecting the optimal model when faced with an *unknown mixture* of tasks, especially with limited or no data to estimate the mixture, a difficult problem. We address this challenge by formulating it as a multiple-source domain adaptation (MSA) problem. We introduce a novel, scalable algorithm that effectively routes each input to the best-suited model from a pool of available models. Our approach provides a strong performance guarantee: remarkably, for any mixture domain, the accuracy achieved by the best source model is maintained. This guarantee is established through a theoretical bound on the regret for new domains, expressed as a convex combination of the best regrets in the source domains, plus a concentration term that diminishes as the amount of source data increases. While our primary contributions are theoretical and algorithmic, we also present empirical results demonstrating the effectiveness of our approach.

## 1 Introduction

Fine-tuning is a key step in adapting large language models (LLMs) to specialized tasks or domains after their general pre-training. In this process, an LLM trained on vast datasets is further trained on smaller, task-specific datasets. As organizations and researchers fine-tune LLMs for tasks like summarization, translation, or customer service, the result is a growing collection of models, each optimized for different tasks.

Routing algorithms are crucial for efficiently managing this diversity of specialized models, by determining which existing model best fits a given input. Recently, various routing algorithms have been proposed [Chen et al., 2023; Wang et al., 2023; Hu et al., 2024; Madaan et al., 2023; Yue et al., 2023; Lee et al., 2023; Shnitzer et al., 2023; Narayanan Hari and Thomson, 2023; Lu et al., 2023], including some with strong theoretical and empirical guarantees [Mao et al., 2023, 2024a,b]. While these routing solutions can perform well when inputs are drawn from clearly defined task distributions, they offer no guarantees for inputs originating from an *unknown mixture* of domain distributions. Building a fine-tuned model for every possible task combination is impractical, so how can routing be designed to handle such mixed-task inputs?

To address this problem, this paper frames model routing as a multiple-source domain adaptation (MSA) problem [Mansour et al., 2008] and derives a principled solution for enhancing robustness and adaptability across diverse and dynamic task distributions. Our approach grounded in strong MSA theory [Mansour et al., 2008, 2012; Hoffman et al., 2021; Cortes et al., 2021c] ensures that our routing model system performs as well as the best individual expert model across *any* task mixture. While our contributions are primarily theoretical and algorithmic, we also provide empirical evidence demonstrating the effectiveness of our methods. Our solution is easily implemented and compatible

39th Conference on Neural Information Processing Systems (NeurIPS 2025).

with existing router training approaches. It enhances existing router training by strategically adjusting task domain weights.

A detailed survey of related work is provided in Appendix A. Here, we briefly emphasize that our setting differs significantly from routing in Mixture-of-Experts (MoE) architectures [Shazeer et al., 2017], where experts are trainable components within a unified network. In contrast, our work focuses on predictive model routing [Shnitzer et al., 2023; Narayanan Hari and Thomson, 2023; Lu et al., 2023; Mao et al., 2023, 2024a,b,c, 2025], where the router selects among a fixed set of independently trained models. The central challenge we tackle is making accurate routing decisions when the target distribution is unknown and assumed to be an arbitrary mixture over source domains.

Our analysis builds on the multiple-source adaptation (MSA) framework, first studied theoretically by Mansour et al. [2008, 2012], with subsequent algorithmic and theoretical advances by Hoffman et al. [2021] and Cortes et al. [2021c,a]. Our goal, however, is not to improve theoretical bounds for domain adaptation per se, nor to boost routing accuracy when the test-time distribution is known in advance. Instead, we offer a practical, theoretically grounded framework for robust model routing under *unknown task mixtures*.

The rest of this paper is organized as follows. In Section 2, we formally define the model routing problem under unknown target distributions and cast it in the MSA framework. Section 3 introduces two algorithms along with their practical implementations. Section 4 presents theoretical analyses of our methods, including regret bounds under natural oracle assumptions. We prove a regret bound for arbitrary target mixtures, showing that the regret is a convex combination of the best source-domain regrets, plus a data-dependent concentration term that vanishes with increasing source data. Finally, Section 5 reports our empirical results, demonstrating the benefits of our min-max optimization approach.

## 2 Problem Formulation

We begin by introducing the model routing problem and then show how it can be framed within the multiple-source adaptation (MSA) framework.

### 2.1 Model Routing

We consider a finite set of generative models, denoted by $\Pi$, where each model $\pi\colon \mathcal{X} \to \Delta(\mathcal{Y})$ maps inputs $\mathcal{X}$ to probability distributions over outputs $\mathcal{Y}$. For example, if $\Pi$ consists of generative language models, $\mathcal{X}$ would represent prompts and $\mathcal{Y}$ their corresponding generations. Additionally, we assume there are $k$ benchmark tasks, $D_1, \ldots, D_k$, where each $D_i$ is a distribution over inputs. Typically, access to $D_i$ is limited to a finite dataset. We will denote by $\hat{D}_i$ the empirical distribution consisting of $n_i$ i.i.d. samples drawn from $D_i$. Let $r^\star\colon \mathcal{X} \times \mathcal{Y} \to [0, 1]$ represent a scoring function that evaluates the quality of a generation $y \in \mathcal{Y}$ for a given input $x \in \mathcal{X}$. For example, $r^\star$ could indicate the probability that human evaluators prefer $y$ over the output of a reference model. Although $r^\star$ may be unknown, we assume access to a scoring oracle $\mathcal{R}$ that provides unbiased estimates of $r^\star$ for any input-output pair $(x, y)$. For simplicity, we assume that the scoring function $r^\star$ is uniform across all benchmark tasks, though this assumption can be relaxed. The *value* of a model $\pi \in \Pi$ on an input $x$ or distribution over inputs $D$ is defined as follows:

$$v(\pi, x) = \mathop{\mathbb{E}}_{y \sim \pi(x)}\big[r^\star(x, y)\big] \qquad v(\pi, D) = \mathop{\mathbb{E}}_{x \sim D}\big[v(\pi, x)\big].$$

**Goal of predictive model routing.** Given access to $\Pi$, $\mathcal{R}$, and the datasets $\widehat{D}_1, \ldots, \widehat{D}_k$, our goal is to select a high-quality probabilistic *routing function* $f\colon \mathcal{X} \to \Delta(\Pi)$ from a family $\mathcal{F}$ of such functions. Each routing function maps an input $x \in \mathcal{X}$ to a probability distribution over the models in $\Pi$. For any input $x$, a model $\pi \in \Pi$ is selected by sampling from the distribution $f(x)$.

For any $\pi\colon \mathcal{X} \to \Delta(\mathcal{Y})$ and $(x, y) \in \mathcal{X} \times \mathcal{Y}$, let $\pi(y|x)$ denote the probability of $y$ under the distribution $\pi(x)$. Given a routing function $f \in \mathcal{F}$, we define the induced distribution $\pi_f(\cdot|x)$ over outputs $\mathcal{Y}$ as:

$$\forall (x, y) \in \mathcal{X} \times \mathcal{Y}, \quad \pi_f(y|x) = \sum_{\pi \in \Pi} f(\pi|x)\pi(y|x).$$

The objective is for $f$ to route inputs $x$, drawn from an unknown test domain $D \in \Delta(\mathcal{X})$, to models in $\Pi$ that yield high scores according to the oracle $\mathcal{R}$. Specifically, we aim to find an $f$ that maximizes

the expected score $v(\pi_f, D)$, without prior knowledge of $D$. The performance of a routing function $f$ is evaluated by the *regret* of its induced policy $\pi_f$ on the test domain $D$, defined as:

$$\mathrm{reg}(\pi_f, D) \coloneqq \max_{\pi \in \Pi} v(\pi, D) - v(\pi_f, D), \tag{1}$$

that is the gap between the performance of the best model in $\Pi$ and that of the model selected by $f$.

**Why is the test domain unknown?** The test distribution $D$, representing the real-world data an application will encounter, is typically unknown during development. We often know which kind of tasks a model may be used for but how popular a specific task is, cannot be known ahead of deployment and may also change over time. We therefore focus on the problem where $D$ is composed of a mixture of the benchmark tasks $D_1, \ldots, D_k$ but the weights of this mixture are unknown at training time of the routing function. We formalize this in the next section.

## 2.2 Predictive Model Routing as Multiple-Source Domain Adaptation

Multiple-source domain adaptation (MSA) is a closely related problem that has been extensively studied, particularly in classification and regression problems [Mansour et al., 2008, 2012; Hoffman et al., 2021; Cortes et al., 2021c]. In MSA, the task involves multiple source domains, $D_1, \ldots, D_k$, each associated with a near-optimal model $h_1, \ldots, h_k$ [Mansour et al., 2008]. The target domain, $D_\lambda$, is defined as a $\lambda$-mixture of the source domains, $D_\lambda = \frac{1}{k} \sum_{i=1}^{k} \lambda_i D_i$, where $\lambda \in \Delta([k])$ represents unknown mixture weights. The objective is to devise a combination rule for the models $h_i$ such that the resulting model performs well on any target domain $D_\lambda$.

We can formulate the predictive model routing problem as a multiple-source domain adaptation task by first selecting an appropriate model, $\pi_i$, for each dataset, which we refer to as the expert model for domain $D_i$. In many applications, natural choices for $\pi_i$ arise, such as when a model $\pi$ has been fine-tuned to perform well on a specific domain $D_i$. More generally, we can define $\pi_i$ as the model in the set $\Pi$ that achieves the highest value estimate for $D_i$. Next, we augment the empirical distributions $\widehat{D}_1, \ldots, \widehat{D}_k$ with score samples from each expert model. For each input $x$ in the support of $\widehat{D}_i$, we compute scores $r_1, \ldots, r_k$ by generating responses $y_j \sim \pi_j(\cdot|x)$ from each expert $\pi_j$ and querying the reward oracle, which returns scores $r_j \sim \mathcal{R}(x, y_j)$. These scores, $r_j$, serve as unbiased estimates of the value $v(\pi_j, x)$. We denote the augmented version of $\widehat{D}_i$ as $\bar{D}_i$.

With the *score-augmented distributions* $(\bar{D}_i)_{i \in [k]}$ in hand, the objective is to find a routing function (or combination rule) $f: \mathcal{X} \to \Delta([k])$ that maps inputs to a distribution over expert models. This routing function induces a mixed generation policy $\pi_f(y|x) = \sum_{i=1}^{k} f(i|x) \pi_i(y|x)$, which is evaluated based on its performance across any target domain $D_\lambda$. The quality of the routing function $f$ is measured by its regret relative to the full policy set $\Pi$, as defined in (1). For the remainder of the paper, we adopt this domain adaptation perspective on predictive model routing, assuming that we are provided with a score-augmented empirical distribution $\bar{D}_i$ for each domain $D_i$ and that the goal is to learn an effective routing function to the expert models.

## 3 Algorithm

To ensure robustness in model routing across test domains, we draw on two key areas of research: multiple-source domain adaptation [Mansour et al., 2008; Cortes et al., 2021c] and minimax-regret optimization [Alaiz-Rodrıguez et al., 2007; Rigter et al., 2021; Mohri et al., 2019; Agarwal and Zhang, 2022]. Our approach is particularly aligned with the approaches of Cortes et al. [2021c], Mohri et al. [2019] and Agarwal and Zhang [2022]. Specifically, we adopt the mixture over test domains and the associated theoretical guarantees from [Cortes et al., 2021c], while the objective formulation and optimization strategy are inspired by [Mohri et al., 2019; Agarwal and Zhang, 2022].

To design our algorithm, we begin by considering the idealized infinite-data setting and then introduce finite-sample approximations. Rather than minimizing regret under a fixed distribution, as defined in (1), we adopt a more robust objective inspired by the minimax regret optimization literature [Alaiz-Rodrıguez et al., 2007; Rigter et al., 2021; Mohri et al., 2019; Agarwal and Zhang, 2022]. Specifically, we aim to *minimize the worst-case regret over all possible test domains*:

$$\min_{f \in \mathcal{F}} \max_{\lambda \in \Delta([k])} \max_{\pi' \in \Pi} v(\pi', D_\lambda) - v(\pi_f, D_\lambda). \tag{2}$$

However, solving this optimization problem during training is challenging due to the maximization over $\pi' \in \Pi$. To address this challenge, we propose two practical variants that avoid optimization over $\pi'$. Each variant minimizes regret relative to a specific policy, denoted as $\pi_A^\star$ or $\pi_B^\star$.

**Option A: Pointwise Comparator.** In this first variant, we aim to compete against a policy $\pi_A^\star$ that, for each input context $x$, achieves the performance of the best expert model. Formally, $v(\pi_A^\star, x) = \max_{i \in [k]} v(\pi_i, x)$ for all $x$. This leads to the following objective:

$$\min_{f \in \mathcal{F}} \max_{\lambda \in \Delta([k])} \mathcal{L}_A(f, \delta) := \min_{f \in \mathcal{F}} \max_{\lambda \in \Delta([k])} v(\pi_A^\star, D_\lambda) - v(\pi_f, D_\lambda). \tag{3}$$

In the finite-sample setting, this min-max objective becomes:

$$\min_{f \in \mathcal{F}} \max_{\lambda \in \Delta([k])} \widehat{\mathcal{L}}_A(f, \delta) := \min_{f \in \mathcal{F}} \max_{\lambda \in \Delta([k])} \mathop{\mathbb{E}}_{\substack{i \sim \lambda \\ (x, r_1, \ldots, r_k) \sim \bar{D}_i}} \left[ \max_{j \in [k]} r_j - \sum_{l=1}^{k} f(l|x) \, r_l \right]. \tag{4}$$

where the maximum is taken over expert scores for each sample. While being easy to implement, this approach introduces additional bias when there is high variance in the expert scores for a given input.

**Option B: Domain Comparator.** To limit bias in the finite-sample objective, we leverage the structure of the model routing problem by using $\pi_B^\star$ as the comparator in the regret calculation. This policy, $\pi_B^\star : \mathcal{X} \times [k] \to \Delta(\mathcal{Y})$, takes both the input $x$ and the domain label $i$, following the expert model $\pi_i$ for samples from domain $D_i$; that is, $\pi_B^\star(x, i) = \pi_i(x)$. As we will demonstrate later, this fixed comparator provides strong regret guarantees without requiring an additional inner optimization over policies. This leads to the following optimization objective:

$$\min_{f \in \mathcal{F}} \max_{\lambda \in \Delta([k])} \mathcal{L}_B(f, \delta) := \min_{f \in \mathcal{F}} \max_{\lambda \in \Delta([k])} v(\pi_B^\star, D_\lambda) - v(\pi_f, D_\lambda) \tag{5}$$

with the finite-sample counterpart:

$$\min_{f \in \mathcal{F}} \max_{\lambda \in \Delta([k])} \widehat{\mathcal{L}}_B(f, \delta) := \min_{f \in \mathcal{F}} \max_{\lambda \in \Delta([k])} \mathop{\mathbb{E}}_{\substack{i \sim \lambda \\ (x, r_1, \ldots, r_k) \sim \bar{D}_i}} \left[ r_i - \sum_{l=1}^{k} f(l|x) \, r_l \right]. \tag{6}$$

Note that $\pi_A^\star$ and $\pi_B^\star$ coincide when domain experts are perfect, producing the best score for each individual $x$ from their respective domain. However, in practice, even $\pi_i$ that are well-tuned for their domain $D_i$ typically do not achieve this, which distinguishes $\pi_A^\star$ from $\pi_B^\star$ in general.

**Algorithm.** We follow the standard approach and tackle the saddle-point problems in Equation 4 or 6 as a two-player game, which can be solved by dueling two no-regret learners (see Mohri et al. [2019] for a general Mirror descent solution). Our algorithm is shown in Algorithm 1. The max-player can be solved efficiently with Hedge [Littlestone and Warmuth, 1994]. For the min-player, we do not prescribe the exact update for $f_t$ as we do not wish to prescribe a specific function class $\mathcal{F}$. Instead, we follow prior work [e.g. Cheng et al., 2022] and rely on online learning oracles. When the update aims to optimize (4) or (6) directly, which are linear losses in the predictions of $f_t$, then we refer to this as an OLO oracle. Alternatively, it is common in practice to frame such an objective as a weighted classification problem and instead aim to minimize the weighted log-loss as a proxy instead. We also support such a choice for updates and refer to it as an OLLO oracle. We formalize the notion of oracle rigorously in the next section, but we generally assume that the oracle of choice is a no-regret learner and note that there is a large family of online-learning algorithms available with appropriate guarantees [Cesa-Bianchi and Lugosi, 2006].

**Practical Implementation.** Algorithm 1 can be seamlessly integrated into existing model training frameworks. For instance, in the case of language model routing, the class $\mathcal{F}$ can be a moderate-sized language model architecture, where the initial policy $f_1$ is a pre-trained model with its final layer replaced by a randomly initialized linear layer. At each round $t \in [T]$, a batch of samples is drawn from the augmented datasets, with equal proportions from each. The Hedge update of domain weights $\lambda_t$ can be efficiently computed in closed form with minimal computational cost.

The update of $f_t$ is handled using standard gradient-based optimizers on the objectives in (4) or (6), augmented with a KL-regularization, similar to RLHF training objectives [Christiano et al., 2017], such as regularization toward a uniform domain distribution or a given domain prior. This corresponds to an OLO oracle. Alternatively, the model can be optimized with a proxy log-loss, similar to standard supervised fine-tuning objectives and corresponding to an OLLO oracle.

**Algorithm 1:** Domain adaptation for model routing algorithm

---

1 **Input:** Score-augmented distributions $\bar{D}_i$ for $i \in [k]$ of size $n_i$. Each sample is of the form
  $(x, r_1, \ldots, r_k)$ where $x$ is the context and $r_j$ is a reward estimate for expert policy $\pi_j$;
2 **Output:** Routing policy $f : \mathcal{X} \to \Delta_k$;
3 Initialize $\lambda_1 = [\frac{1}{k}, \ldots, \frac{1}{k}]^\top$ and $f_1$ in $\mathcal{F}$ arbitrarily;
4 **for** $t = 1, 2, \ldots, T$ **do**
5     Sample $(x_t^{(i)}, r_{t,1}^{(i)}, \ldots, r_{t,k}^{(i)}) \sim \bar{D}_i$ for each $i \in [k]$;
6     Determine benchmark with **option A** $y_t^{(i)} \in \mathrm{argmax}_{j \in k} \, r_{t,j}^{(i)}$ or **option B** $y_t^{(i)} = i$;
7     Set benchmark score $c_t^{(i)} = r_{t,y_t^{(i)}}^{(i)}$ ;
8     **Max-player: Hedge**
9     Update $\lambda_{t+1} \propto \lambda_t \exp(-\gamma \ell_t)$ with losses $\ell_t \in \mathbb{R}^k$ where $\ell_{t,i} = c_t^{(i)} - \sum_{j=1}^k r_{t,j}^{(i)} f_t(j | x_t^{(i)})$.
10     **Min-player: no-regret online learning update**
11     Update $f_{t+1}$ with $\mathsf{OLO}$ for all $i \in [k]$ contexts $x_t^{(i)}$ and losses $\ell_t^{(i)} \in \mathbb{R}^k$ with
  $\ell_{t,j}^{(i)} = \lambda_{t,i} (c_t^{(i)} - r_{t,j}^{(i)})$
12          or $\mathsf{OLLO}$ for all $i \in [k]$ contexts $x_t^{(i)}$ and targets $y_t^{(i)}$ with weight $\lambda_{t,i}$;
13 **return** $\bar{f} = \frac{1}{T} \sum_{t=1}^T f_t$

---

Finally, Algorithm 1 returns an averaged model $\bar{f}$, where $\bar{f}(i|x) = \frac{1}{T} \sum_{t=1}^T f_t(i|x)$ for all $x \in \mathcal{X}$ and $i \in [k]$. While exact output averaging might not always be feasible, we can adopt a "model souping" approach by averaging the parameters $\theta_t$ of the models $f_t$ across iterations. The final model is then represented by $\bar{\theta} = \frac{1}{T} \sum_{t=1}^T \theta_t$, a technique that has proven effective in practice [Wortsman et al., 2022; Ramé et al., 2024].

## 4 Theoretical Analysis

Before formally stating our results, we first introduce the oracle assumptions.

### 4.1 Oracles

The first oracle we support is a no-regret learner directly on the losses linear in $f$:

**Definition 1** (Online learning oracle). *An algorithm $\mathsf{OLO}$ is referred to as an online learning oracle for a class $\mathcal{F} \subseteq \mathcal{X} \to \Delta_k$ if it satisfies the following condition. Given an arbitrary, potentially adversarial sequence of context-loss pairs $(x_1, \ell_1, \ldots, x_{kT}, \ell_{kT})$, $\mathsf{OLO}$ observes the context-loss pairs sequentially and maintains a sequence of policies $f_{t+1} \in \mathcal{F}$, updating the policy after observing $k$ contexts $x_{kt}, \ldots x_{kt+k-1}$ and losses $\ell_{kt}, \ldots, \ell_{kt+k-1}$. The regret of $\mathsf{OLO}$ is given by:*

$$\mathrm{Reg}_{\mathcal{F}}^{\mathsf{OLO}}(T) = \max_{f \in \mathcal{F}} \sum_{t=1}^T \sum_{i=0}^{k-1} \langle f(x_{kt+i}) - f_t(x_{kt+i}), \ell_{kt+i} \rangle,$$

*and is sublinear $\mathrm{Reg}_{\mathcal{F}}^{\mathsf{OLO}}(T) = o(T)$ with probability at least $1 - \delta$.*

We expect that standard gradient-based optimizers on the objectives in (4) or (6), augmented with appropriate entropy regularization, are no-regret learners that satisfy this definition at least approximately. Note that our definition of online learning oracle deviates slightly from the standard no-regret definition because it uses batched updates where $f_t$ is only updated after $k$ predictions. This is a minor difference and algorithms that satisfy the standard definition also satisfy our definition above with regret increased by a factor of at most $k$.

For the alternate updates of $f_t$ with $\mathsf{OLLO}$, we view the problem as a classification problem with weighted datapoints and prescribe that the updates lead to small log-loss (or cross-entropy loss), formalized as:

**Definition 2** (Online log-loss oracle). *An algorithm $\mathsf{OLLO}$ is referred to as an online log-loss oracle for a class $\mathcal{F} \subseteq \mathcal{X} \to \Delta_k$ if it satisfies the following condition. Given an arbitrary, potentially*

| Oracle | Option | Main term | Assumptions |
|--------|--------|-----------|-------------|
| OLO | A | $\text{reg}(\pi_A^\star, D_\lambda)$ | best expert per $x$ deterministic & $\mathcal{F}$ contains perfect predictor for best score on $\widehat{D}_1, \ldots, \widehat{D}_k$ |
| OLO | A | $\text{reg}(\pi_A^\star, D_\lambda) + \text{bias}_A + O(\sqrt{1/n_{\min}})$ | & $\mathcal{F}$ contains perfect predictor for best expected score on $D_1, \ldots, D_k$ |
| OLO | B | $\sum_{i=1}^k \lambda_i \text{reg}(\pi_i, D_i)$ | $\mathcal{F}$ contains $f_{\lambda, \widehat{D}}$ for every $\lambda \in \Delta_k$ |
| OLO | B | $\sum_{i=1}^k \lambda_i \text{reg}(\pi_i, D_i) + O(\sqrt{1/n_{\min}})$ | $\mathcal{F}$ contains $f_{\lambda, D}$ for every $\lambda \in \Delta_k$ |
| OLLO | A & B | $\min_{f \in \mathcal{F}} \max_{i \in [k]} \mathbb{E}_{x^{(i)} \sim \bar{D}_i}[-\log(f(y^{(i)}|x^{(i)}))]$ | second moment of logits bounded for all $f \in \mathcal{F}$ |

Table 1: Summary of Guarantees. $n_{\min} = \min_{i \in [k]} n_i$ denotes the size of the smallest dataset.

*adversarial sequence of context-weight-target triples* $(x_1, w_1, y_1, w_1, \ldots, x_{kT}, w_{kT}, y_{kT})$, *OLLO observes the triples sequentially and maintains a sequence of policies* $f_{t+1} \in \mathcal{F}$, *updating the policy after observing* $k$ *contexts* $x_{kt}, \ldots x_{kt+k-1}$, *weights* $w_{kt}, \ldots w_{kt+k-1}$ *and targets* $y_{kt}, \ldots, y_{kt+k-1}$. *The regret of* OLO *is given by:*

$$\text{Reg}_{\mathcal{F}}^{OLLO}(T) = \max_{f \in \mathcal{F}} \sum_{t=1}^{T} \sum_{i=0}^{k-1} w_{kt+i} \log\left(\frac{f_t(y_{kt+i}|x_{kt+i})}{f(y_{kt+i}|x_{kt+i})}\right)$$

*and is sublinear* $\text{Reg}_{\mathcal{F}}^{OLLO}(T) = o(T)$ *with prob.* $\geq 1 - \delta$.

## 4.2 Guarantees

We begin by building intuition for the terms that will appear in our final error bounds. Since we do not have access to the underlying domains $D_i$ directly, but only through datasets of finite size, we should expect to pay for a finite-sample error for this approximation. For a test domain $D_\lambda$, this appears in our bounds as a concentration term of the form

$$\text{Conc}(\lambda, n) = O\left(\sqrt{\sum_{i=1}^{k} \frac{\lambda_i^2 \log\left(\frac{\mathcal{C}_\mathcal{F}}{\delta}\right)}{n_i}} + \max_i \frac{\lambda_i \log\left(\frac{\mathcal{C}_\mathcal{F}}{\delta}\right)}{n_i}\right),$$

where $\mathcal{C}_\mathcal{F}$ is an appropriate complexity measure for $\mathcal{F}$, e.g. the $\ell_\infty$ covering number of $\mathcal{F}$. The error term $\text{Conc}(\lambda, n)$ decreases as the sizes of the datasets $n_i$ increase. Fortunately, $\lambda$ here is the mixture weights of the test domain and thus, the dominating first term that shrinks slowest with $n_i$ only depends on the dataset size of source domain that build the support of $D_\lambda$.

Another source of error in [Algorithm 1](#) is that it aims to find solutions to [(6)](#) or [(4)](#) in an iterative fashion and if the number of iterations is small, we incur additional approximation errors. This yields error terms of the form

$$\text{Approx}(T) = \frac{\text{Reg}_\mathcal{F}(T)}{T} + O\left(\sqrt{\frac{\log(k\mathcal{C}_\mathcal{F}/\delta)}{T}}\right)$$

where $\text{Reg}_\mathcal{F}(T)$ is either $\text{Reg}_\mathcal{F}^{OLO}(T)$ or $\text{Reg}_\mathcal{F}^{OLLO}(T)$, depending on the oracle choice. This term includes the error due to updates of $f_t$ (hence the $\text{Reg}_\mathcal{F}(T)$ dependency), error due to updating $\lambda_t$ (regret of Hedge) and uniform concentration errors due to sampling datapoints for updates from the dataset in each iteration. As long as the chosen oracle work and $\text{Reg}_\mathcal{F}(T) = o(T)$, this $\text{Approx}(T)$ term will shrink with $T$ and we can make it as close to zero as we like by running [Algorithm 1](#) for enough iterations. Note that increasing the number of iterations $T$ only incurs a computational overhead but does not require more data.

We can now state our main result when $f_t$-updates are performed by an OLO oracle. We discuss here the results for OLO oracles and defer those for OLLO oracles to the appendix (see [Table 1](#)).

**Theorem 1.** *Let* $D_\lambda$ *be the test domain,* $\mathcal{F}$ *be a convex set and updated to* $f_t$ *performed by an* OLO *oracle. Then, with probability at least* $1 - O(\delta)$, *the regret of the function* $\bar{f}$ *returned by [Algorithm 1](#) with Option A satisfies*

$$\text{reg}(\pi_{\bar{f}}, D_\lambda) \leq \text{reg}(\pi_A^\star, D_\lambda) + \widehat{V}_A^\star + \text{Approx}(T) + \text{Conc}(\lambda, n),$$

where $\pi_A^\star$ is the competitor policy for option A and $\widehat{V}_A^\star = \max_{\lambda \in \Delta_k} \inf_{f \in \mathcal{F}} \widehat{\mathcal{L}}(f, \lambda)$ is the optimal value of the objective in Equation 4. The same guarantee holds for Option B with $\pi_A^\star$ replaced by $\pi_B^\star$ and $\widehat{V}_A^\star$ by $\widehat{V}_B^\star$.

In addition to the two error terms $\mathrm{Approx}(T)$ and $\mathrm{Conc}(\lambda, n)$ discussed above, our performance guarantee contains the main term $\mathrm{reg}(\pi_A^\star, D_\lambda) + \widehat{V}_A^\star$. Since the game in (5) and (3) are designed to make $\pi_f$ match the performance of the chosen competitor $\pi_A^\star$ or $\pi_B^\star$, the regret of this competitor naturally appears in the our error bound. The $\widehat{V}_A^\star$ term is the value of the game / saddle-point and small as long as the function class $\mathcal{F}$ is expressive enough. How expressive $\mathcal{F}$ needs to be depends on the chosen option.

**Option B.** Building on the analysis of Mansour et al. [2008], we show in Lemma 7 in the appendix that the value $\widehat{V}_B^\star$ is non-positive, as long as $\mathcal{F}$ can represent the conditional probabilities of a datapoint coming from each source dataset given its context $x$ which, by Bayes rule, is

$$f_{\lambda, \widehat{D}}(i|x) = \frac{\lambda_i \widehat{D}_i(x)}{\sum_{j=1}^k \lambda_j \widehat{D}_j(x)}.$$

Further note that due to the specific definition of $\pi_B^\star$, its regret can be bounded by the regret of each domain expert on its own domain

$$\mathrm{reg}(\pi_B^\star, Q_\lambda) = \max_{\pi' \in \Pi} v(\pi', D_\lambda) - \sum_{i=1}^k \lambda_i v(\pi_i, D_i) \leq \sum_{i=1}^k \lambda_i \mathrm{reg}(\pi_i, D_i).$$

This yields the following corollary of Theorem 1:

**Corollary 1.** *Assume $\mathcal{F}$ to be a convex set and assume $\mathcal{F}$ contains $f_{\lambda, \widehat{D}}$ for every $\lambda \in \Delta_k$. Let $D_\lambda$ be the test domain. Then, with probability at least $1 - O(\delta)$ the regret $\mathrm{reg}(\pi_{\bar{f}}, D_\lambda)$ of $\bar{f}$ returned by Algorithm 1 with OLO oracle and Option B is bounded by*

$$\sum_{i=1}^k \lambda_i \mathrm{reg}(\pi_i, D_i) + \mathrm{Approx}(T) + \mathrm{Conc}(\lambda, n).$$

This recovers the canonical results from prior domain adaptation works, and shows that as long as the experts are optimal on their domain, the produced routed model will be optimal on any test domain $D_\lambda$ as well , up to the error terms which decrease with $T$ and $n$.

We expect that for many modern neural network architectures, the condition that $\mathcal{F}$ can represent $f_{\lambda, \widehat{D}}$ to hold. However, this condition does depend on the (randomly drawn) datasets $\widehat{D}_i$. To obtain a condition that does not depend on the realized datasets, we can instead just assume that the condition holds for the true source domains $D_i$. However, we incur an additional $O(\sqrt{k/\min_i n_i})$ error term in this case. See Table 1 for a summary and the appendix for details.

Further, we expect the algorithm performance to degrade gracefully when the realizability assumption in Corollary 1 does not hold exactly and that the bound could be extended to reflect this through an additional additive approximation error scaling with $\min_{f \in \mathcal{F}} \max_{i \in [k]} \mathbb{E}_{x \sim \hat{D}_i} \left[ |f(\cdot|x) - f_{\lambda, \hat{D}}(\cdot|x)|_1 \right]$.

**Computational efficiency at inference time** We note that when deploying the router $\bar{f}$ returned by Algorithm 1 there is no need to call all of the domain experts $\pi_i$. Instead, given a context, we sample $i \sim \bar{f}(\cdot|x)$ and only play according to $\pi_i(\cdot|x)$. This modification inherits the regret guarantees of Theorem 1 as the sampled $y$ follows exactly the same distribution as that of $y \sim \pi_f(\cdot|x)$ and so the values of the two sampling procedures are exactly the same. In Figure 2 we show an experiment which shows the regret with respect to the above sampling for Option A with log-loss. We also include a comparison in Table 2.

**Option A.** To show that $\widehat{V}_A^\star$ is small, we need stronger assumptions compared to Option B. In order for $\widehat{V}_A^\star$ to be zero, there must be a $f \in \mathcal{F}$ that can match the performance of $\pi_A^\star$ which always takes the best expert for a given sample. This $f$ needs to satisfy

$$\mathbb{E}_{(x, r_1, \dots r_k) \sim \bar{D}_i} \left[ \max_j r_j - \sum_{m=1}^k f(m|x) r_m \right] = 0 \quad \forall i. \tag{7}$$

| Loss for $f$ | Option | regret vs best expert | | regret vs domain expert | |
|---|---|---|---|---|---|
| | | Baseline | Alg 1 | Baseline | Alg 1 |
| linear | A | 4.60 | **4.28** | 1.65 | **0.49** |
| linear | B | **4.60** | 7.09 | 1.64 | **1.08** |
| log | A | 2.70 | **2.37** | -0.06 | **-0.39** |
| log (sampled) | A | 2.70 | **1.90** | -0.06 | **-0.85** |
| log | B | 7.90 | **7.84** | 0.58 | **0.23** |

Table 2: Overview of regret in the worst-case test domain (lower is better), comparing the routing function produced by Algorithm 1 against a routing function produced by training with uniform and fixed domain weights. Results are averages across 5 seeds. Algorithm 1 consistently reduces the regret against the competitor targeted by the selected option.

If there is significant stochasticity in the rewards $r_i$ generated by an expert for a given $x$, then this condition might be impossible to satisfy. Specifically, if there are multiple samples with the same $x$ but where optimal score is obtained by different experts with distinct scores, then no such mapping can exist. However, a sufficient condition for such an $f$ to exist is that for every $x$ there is some expert which with probability 1 always has the highest score among all experts. In this case, the finite-sample approximation in (4) is an unbiased estimator of (3). If, in addition this $f$ is contained in $\mathcal{F}$, then we can show that $\widehat{V}_A^\star = 0$ and the regret bound for $\pi_{\bar{f}}$ becomes

$$\mathrm{reg}(\pi_{\bar{f}}, D_\lambda) \le \mathrm{reg}(\pi_A^\star, D_\lambda) + \mathrm{Approx}(T) + \mathrm{Conc}(\lambda, n).$$

In general, if there are contexts for which the distributions of scores between experts overlap and the identity of the best expert varies with the random draws of generations and score assignment, then (4) incurs a bias and we can still show $\widehat{V}_A^\star \le \mathrm{bias}_A + O\left(\frac{k \log(\mathcal{C}_\mathcal{F}/\delta)}{\sqrt{\min_i n_i}}\right)$ where we incur an additional $1/\sqrt{\min_i n_i}$ term and the bias

$$\mathrm{bias}_A = \max_i \mathop{\mathbb{E}}_{x \sim \hat{D}_i} \left[ \mathop{\mathbb{E}}_{r_{1:k}|x} \left[\max_m r_m\right] - \max_m v(\pi_m, x) \right].$$

This term essentially quantifies the bias from estimating the maximum expected score by taking the maximum over realizations, thus exchanging the expectation and maximum in the expression. There are many scenarios where $\mathrm{bias}_A$ is favorably small. For example, we can show that for any $x$

$$\mathop{\mathbb{E}}_{r_{1:k}|x} \left[\max_m r_m\right] - \max_m v(\pi_m, x) \le \sqrt{k \max_i \sigma_i^2}$$

holds where $\sigma_i^2$ is the variance of $r_i$ and thus when all scores are near-deterministic $\mathrm{bias}_A$ is small. Another favorable case is when for every $x$ there is some model that is always the best, in which case $\mathrm{bias}_A = 0$.

**Option A vs Option B.** By construction, $\pi_A^\star$ is a stronger competitor than $\pi_B^\star$, since the inequality $v(\pi_B^\star, x) \le \max_{i \in [k]} v(\pi_i, x) = v(\pi_A^\star, x)$ holds for all $x \in \mathcal{X}$. Thus, in the most favorable cases when $\bar{V}_A^\star = 0$, Option A is indeed preferable as $\mathrm{reg}(\pi_A^\star, D_\lambda) \le \sum_{i=1}^k \lambda_i \mathrm{reg}(\pi_i, D_i)$. However, as we alluded to earlier, the conditions for $\widehat{V}_A^\star = 0$ are much stronger than those required for $\widehat{V}_B^\star$. Thus, in many practical settings, Option B might be preferable to Option A.

## 5 Empirical Evaluation

Our primary contributions are theoretical and algorithmic, but we now also validate the effectiveness of our approach empirically. Prior studies have explored optimal strategies for learning a routing function tailored to specific data distributions [e.g. Jiang et al., 2023; Hu et al., 2024]. We view our algorithm as a framework that can be applied on top of these approaches, by using them as an oracle for updating $f_t$. Thus, we do not aim to compare different learning methodologies but to assess the impact of applying our framework to a given oracle choice, and how it makes the routing function more robust by adjusting the domains weights in training. We therefore compare Algorithm 1 with and without updates to $\lambda_t$ (i.e., $\gamma = 0$ vs. $\gamma \ne 0$), while keeping other parameters fixed.

As oracles, we use stochastic gradient based updates on either the linear objective in (6) or (4) directly or on a log-loss proxy, as described in Definition 2. The routing function $f$ is initialized as a

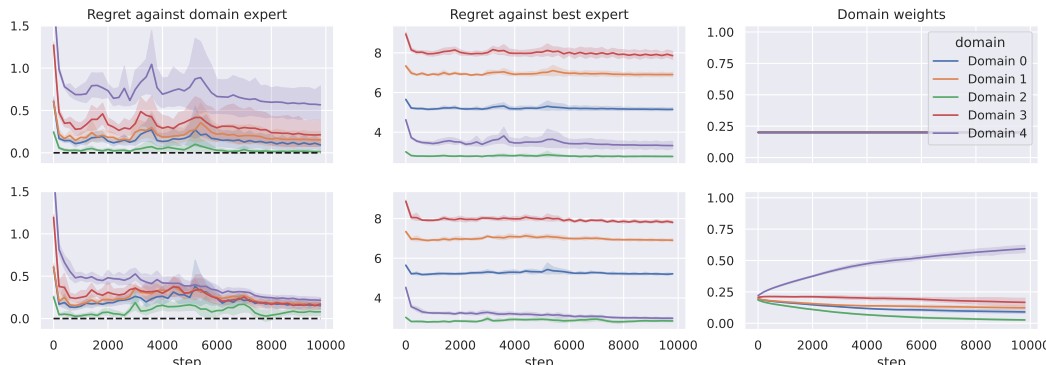

Figure 1: Comparison of Algorithm 1 with Option B (bottom row) versus the baseline without domain weight adjustment (top row), evaluated using log-loss under an OLLO oracle. The left panel shows the regret against each expert, the middle one the regret against the best per example expert.

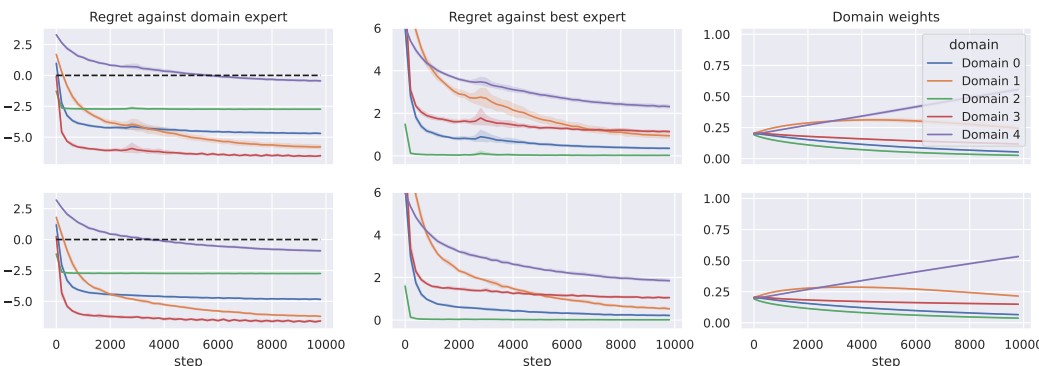

Figure 2: Comparison of Algorithm 1 with Option A (bottom) where rewards are sampled according to the computationally efficient inference procedure against Algorithm 1 with Option A without sampling (top), corresponding to an OLLO oracle.

pre-trained Gemma 2B model [Team et al., 2024], with the final layer replaced by a fully connected, randomly initialized linear layer to produce the logits of $f$. When we use Algorithm 1 with Option A, we use a learning rate $\gamma = 1e-4$ and with Option B a learning rate $\gamma = 1e-3$ to update the domain weights. We choose different learning rates as the magnitude of regret for the max-player is different due the the different comparator in the regret. We train each routing function for 10,000 batches of 256 samples from each domain. Each experiment is repeated 5 times where the datasets are shuffled between instances of the same experiment.

We conduct our evaluation on the MixInstruct benchmark by Jiang et al. [2023] which consists of 5 individual domains. Each domain $\widehat{D}_i$ contains samples with prompts and various metrics for the generations of 11 open-source LLMs. We focus exclusively on the BLEU score and select the model with the highest average BLEU score per domain from the training split to serve as the domain expert.

Table 2 shows a summary of our empirical results, listing for each choice of oracle (loss function linear or log-loss) and algorithm option (A or B) the average regret of our algorithm per step against the baseline which does not update domain weights during training on the the worst-case domain mixture. Algorithm 1 reduces in all cases the regret targeted by the chosen option, regret against best expert for Option A and regret against domain expert for Option B. While this typically results in a reduction of the other notion of regret, an increase may happen (e.g., for linear loss and Option B). The regret is computed with respect to $\pi_f$, that is with respect to $\sum_{j=1}^{k} f_t(i|x_t^{(i)}) r_{t,j}^{(i)}$, except for log (sampled), where we sample $j \sim f_t(\cdot|x_t^{(i)})$ and play the corresponding $r_{t,j}^{(i)}$. Overall, the table shows that the efficient inference strategy performs on par with, or better than, computing regret using the full distribution $f_t(\cdot \mid x_t^{(i)})$.

Figure 1 shows the regret of the routing model on each domain during training, for the example of log-loss and Option B. We show both the regret against each of the domain experts in the first column and the regret against the best per example expert in the second column. The baseline experiment is in the top row and Algorithm 1 with Option B and OLLO is in the bottom row. As expected from the min-max game formulation we see that the regrets against each of the domain experts across domains are being equalized by Algorithm 1. This leads to a decreased regret on Domain 4, compared to the baseline, a regret which matches that of the baseline on domains 1 and 3, and a slightly increased regret for Domain 4 and Domain 0. This is in accordance with how the domain weights, $\lambda_t \in \Delta_k$, change over the course of the game, that is high regret on Domain 4 is penalized more compared to the baseline uniform distribution and the regret of Domain 0 and Domain 2 is penalized less. The second column of Figure 1 shows that, for the harder objective of competing against the per-example best, Algorithm 1 decreases the regret of Domain 4 while maintaining performance on the others. Figure 2 compares the efficient inference strategy with against using the full distribution defined by $f_t$ on our best performing setting for Algorithm 1. As expected, the sampling strategy performs similarly both with respect to the regret and domain weight behavior.

In Appendix D we present results for the remaining 3 settings of Algorithm 1 with Option A and Option B with OLO oracle and with Option A with OLLO oracle. Overall we find the results to be consistent with our observations for Figure 1 and that there is always a benefit to using the min-max optimization approach compared to the uniform weights of the baseline.

## 6   Conclusion

We presented a novel approach for combining multiple domain expert algorithms through the use of online learning oracles, achieving regret bounds that are tightly linked to the performance of these oracles. Our method offers strong theoretical guarantees, ensuring robustness across a broad range of scenarios, including settings with unknown mixtures of source domains. Empirically, we validated the effectiveness of our model routing strategy on the MixInstruct dataset, demonstrating its practical advantages in real-world tasks. These results underscore the promise of our approach as a principled and scalable solution for predictive model selection in heterogeneous environments. Appendix B briefly discusses further extensions of our results.

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

# Contents of Appendix

# A    Related Work

**Mixture-of-experts vs. model routing.** The term *model routing* has been used in the literature to describe related but distinct problems, including token-level routing in Mixture-of-Experts (MoE) architectures [Shazeer et al., 2017]. In MoEs, individual tokens are routed, at each routing layer within a transformer network, to a small subset of expert modules, typically feed-forward networks. These routing layers are trained jointly with the experts, and key challenges include maintaining diversity and ensuring balanced utilization across experts. In contrast, the routing problem we consider in this paper involves assigning the entire input, rather than individual tokens, to one of several fixed (pre-trained) expert models, each specialized for a particular task. When we refer to model routing in this work, we mean this form of input-level assignment. In this setting, load balancing and diversity are not primary concerns.

**Model routing.** Learning a routing function for a collection of fixed specialized models has been studied in the literature in several variations. We can categorize existing work based on the input of the routing function: In post-hoc routing [Chen et al., 2023; Wang et al., 2023; Hu et al., 2024; Madaan et al., 2023; Yue et al., 2023; Lee et al., 2023], the inputs are first processed by the experts and the routing function and selects the best output after seeing both the input and the output from each expert. A specific form of post-hoc routing, known as cascading routing, was studied by Chen et al. [2023]; Wang et al. [2023]; Yue et al. [2023]; Hu et al. [2024], where inputs are processed sequentially by experts until a sufficiently high-quality response is obtained. Theoretical investigations of cascading ideas in classification have been conducted by DeSalvo et al. [2015]. Relatedly, Mao et al. [2023, 2024a,b] have introduced deferral algorithms, which can be used in particular for routing applications, together with an extensive theoretical guarantees. In predictive routing [Shnitzer et al., 2023; Narayanan Hari and Thomson, 2023; Lu et al., 2023] the router only gets to see the input and selects the expert, which alone processes it. Recent efforts by Hu et al. [2024] and Jiang et al. [2023] have proposed benchmarks for evaluating mixtures of LLMs. For a more comprehensive review of this literature, we refer readers to Hu et al. [2024]. Our work focuses on the predictive routing setting, but our techniques can also be applied to other forms of model routing, such as post-hoc routing.

**Multiple-source domain adaptation.** The multiple-source adaptation (MSA) problem was theoretically studied by Mansour et al. [2008, 2012]. Later, Hoffman et al. [2021] introduced an efficient algorithm based on domain density estimation. This approach was subsequently improved by Cortes et al. [2021c], who replaced density estimation with a domain classifier. However, despite this simplification, their method still requires solving a difference of convex (DC) programming problem, which may not be well-suited for modern LLM inference scenarios.

# B    Extensions to other settings

For the sake of exposition, we focus in this paper on predictive routing for generative models. However, our approach can be readily extended to other settings as well. For example, our algorithm can be used without modifications with discriminative base models $\pi$, e.g. for classifying inputs $x$. In that case $\mathcal{Y}$ corresponds to class labels and policies output class probabilities. When we choose scores as binary indicator for a correct classification, our regret guarantees below naturally provide a bound on the misclassification rate of the resulting classifier $\pi_f$. Further, we can tackle post-hoc routing problems, where the decision which model to use is done after seeing the generations from the candidate models, by a small modification of the framework above: Instead of routing functions $f : \mathcal{X} \to \Delta([k])$, we consider functions $f : \mathcal{X} \times \mathcal{Y}^k \to \Delta([k])$ that additionally receive the generations from all candidates as input. In practice this simply corresponds to appending the generations to the prompt $x$ with appropriate divider tokens. Up to passing in this additional information to the routing models, our algorithm remains unaffected and the theoretical guarantees continue to hold.

# C    Theoretical Analysis

Our main performance guarantees for Algorithm 1 with OLO stated in Theorem 1 in the main paper are proven separately for each option in Theorem 2 and Theorem 3 below. The guarantee with OLLO is proven in Theorem 4 afterwards in Appendix C.3

We first derive a set of useful results that apply to all variants. Recall the definitions of the objectives used by our algorithms as

$$\mathcal{L}_A(f,\lambda) = \underset{i\sim\lambda}{\mathbb{E}}\ \underset{x\sim D_i}{\mathbb{E}}\ \underset{j\sim f(x)}{\mathbb{E}}\left[\max_m v(\pi_m,x) - v(\pi_j,x)\right] \tag{8}$$

$$\widehat{\mathcal{L}}_A(f,\lambda) = \underset{i\sim\lambda}{\mathbb{E}}\ \underset{(x,r_1,\ldots,r_k)\sim\widehat{D}_i}{\mathbb{E}}\ \underset{j\sim f(x)}{\mathbb{E}}\left[\max_m r_m - r_j\right] \tag{9}$$

$$\mathcal{L}_B(f,\lambda) = \underset{i\sim\lambda}{\mathbb{E}}\ \underset{x\sim D_i}{\mathbb{E}}\ \underset{j\sim f(x)}{\mathbb{E}}\left[v(\pi_i,x) - v(\pi_j,x)\right] \tag{10}$$

$$\widehat{\mathcal{L}}_B(f,\lambda) = \underset{i\sim\lambda}{\mathbb{E}}\ \underset{(x,r_1,\ldots,r_k)\sim\widehat{D}_i}{\mathbb{E}}\ \underset{j\sim f(x)}{\mathbb{E}}\left[r_i - r_j\right]. \tag{11}$$

In the following, we refer by $\mathcal{L}$ jointly to $\mathcal{L}_A$ or $\mathcal{L}_B$ and $\widehat{\mathcal{L}}$ to $\widehat{\mathcal{L}}_A$ or $\widehat{\mathcal{L}}_B$ respectively.

**Lemma 1.** *The objectives $\mathcal{L}_A, \mathcal{L}_B, \widehat{\mathcal{L}}_A, \widehat{\mathcal{L}}_B$ are bilinear in $f$ and $\lambda$. If $\mathcal{F} \subseteq \mathcal{X} \to \Delta_k$ is convex, then*

$$\inf_{f\in\mathcal{F}} \max_{\lambda\in\Delta_k} \mathcal{L}_A(f,\lambda) = \max_{\lambda\in\Delta_k} \inf_{f\in\mathcal{F}} \mathcal{L}_A(f,\lambda). \tag{12}$$

*and analogously for $\mathcal{L}_B, \widehat{\mathcal{L}}_A$ and $\widehat{\mathcal{L}}_B$.*

*Proof.* We see directly from (8) that all objectives are linear in both arguments. The second part follows from Sion's minimax theorem, since both $\Delta_k$ and $\mathcal{F}$ are convex and $\Delta_k$ is compact. $\square$

The following lemma shows that the costs and rewards concentrate around their expectations.

**Lemma 2.** *The following hold*

$$\mathbb{P}\left(\sup_{\lambda\in\Delta(k)} \sum_{t=1}^T \sum_{i=1}^k \lambda_i\left(c_t^{(i)} - \mathbb{E}[c_t^{(i)}] - \sum_{j=1}^k f_t(j|x_t^{(i)})(r_{t,j}^{(i)} - \mathbb{E}[r_{t,j}^{(i)}])\right) \geq 2\sqrt{T\log(k/\delta)}\right) \leq \delta$$

$$\mathbb{P}\left(\sup_{f\in\mathcal{F}} \sum_{t=1}^T \sum_{i=1}^k \lambda_{t,i}\left(c_t^{(i)} - \mathbb{E}[c_t^{(i)}] - \sum_{j=1}^k f(j|x_t^{(i)})(r_{t,j}^{(i)} - \mathbb{E}[r_{t,j}^{(i)}])\right) \geq 2\sqrt{T\log(|\mathcal{F}|/\delta)}\right) \leq \delta.$$

*Proof.* We start by showing the first inequality. First note that for every $i \in [k]$, $\{c_t^{(i)} - \mathbb{E}[c_t^{(i)}] - \sum_{j=1}^k f_t(j|x_t^{(i)})(r_{t,j}^{(i)} - \mathbb{E}[r_{t,j}^{(i)}])\}_{t\in[T]}$ is a martingale difference sequence with respect to the filtration created by the online oracle. Azuma-Hoeffding's inequality and a union bound implies that

$$\mathbb{P}\left(\sup_{i\in[k]} \sum_{t=1}^T \left(c_t^{(i)} - \mathbb{E}[c_t^{(i)}] - \sum_{j=1}^k f_t(j|x_t^{(i)})(r_{t,j}^{(i)} - \mathbb{E}[r_{t,j}^{(i)}])\right) \geq 2\sqrt{T\log(k/\delta)}\right) \leq \delta.$$

Next, we have

$$\sup_{\lambda\in\Delta(k)} \sum_{i=1}^k \lambda_i \sum_{t=1}^T \left(c_t^{(i)} - \mathbb{E}[c_t^{(i)}] - \sum_{j=1}^k f_t(j|x_t^{(i)})(r_{t,j}^{(i)} - \mathbb{E}[r_{t,j}^{(i)}])\right)$$

$$= \sup_{i\in[k]} \sum_{t=1}^T \left(\mathbb{E}[c_t^{(i)}] - c_t^{(i)} - \sum_{j=1}^k f_t(j|x_t^{(i)})(\mathbb{E}[r_{t,j}^{(i)}] - r_{t,j}^{(i)})\right),$$

since $\sum_{i=1}^k \lambda_i \sum_{t=1}^T \left(c_t^{(i)} - \mathbb{E}[c_t^{(i)}] - \sum_{j=1}^k f_t(j|x_t^{(i)})(r_{t,j}^{(i)} - \mathbb{E}[r_{t,j}^{(i)}])\right)$ is linear in $\lambda$ and the supremum will be achieved at one of the corners of the probability simplex.

The second inequality holds in a similar way by using Azuma-Hoeffding's inequality and a union bound over $\mathcal{F}$. $\square$

We note that the notation $\log(|\mathcal{F}|)$ is overloaded to mean the metric entropy for function classes which have infinite cardinality. For the rest of the paper we consider $\log(|\mathcal{F}|)$ to be the metric entropy with respect to the following distance $d(f, f') = \sup_{x\in\mathcal{X}} \|f(x) - f'(x)\|_1$.

**Lemma 3.** *Let* $\bar{f} = \frac{1}{T}\sum_{t=1}^{T} f_t, \bar{\lambda} = \frac{1}{T}\sum_{t=1}^{T}\lambda_t$ *be the average iterates of Algorithm 1. Then*

$$\max_{\lambda\in\Delta_k, f\in\mathcal{F}}[\widehat{\mathcal{L}}(\bar{f},\lambda) - \widehat{\mathcal{L}}(f,\bar{\lambda})] \leq \frac{\mathrm{Reg}_{\mathcal{F}}(T)}{T} + O\left(\sqrt{\frac{\log(k|\mathcal{F}|/\delta)}{T}}\right) \tag{13}$$

*with high probability at least* $1 - O(\delta)$*, where* $\mathrm{Reg}_{\mathcal{F}}(T)$ *is the regret of the online learning oracle from Definition 1.*

*Proof.* We begin by noting that

$$\widehat{\mathcal{L}}(\bar{f},\lambda) = \mathbb{E}_{i\sim\lambda, x^{(i)}\sim\bar{D}_i}\left[\sum_{i=1}^{k}\lambda_i(c^{(i)} - \langle\bar{f}, r^{(i)}\rangle)\right] = \frac{1}{T}\sum_{t=1}^{T}\sum_{i=1}^{k}\lambda_i\left(\mathbb{E}[c_t^{(i)}] - \sum_{j=1}^{k} f_t(j|x_t^{(i)})\,\mathbb{E}[r_{t,j}^{(i)}]\right)$$

$$\widehat{\mathcal{L}}(f,\bar{\lambda}) = \mathbb{E}_{i\sim\lambda, x^{(i)}\sim\bar{D}_i}\left[\sum_{i=1}^{k}\bar{\lambda}_i(c^{(i)} - \langle f, r^{(i)}\rangle)\right] = \frac{1}{T}\sum_{t=1}^{T}\sum_{i=1}^{k}\lambda_{t,i}\left(\mathbb{E}[c_t^{(i)}] - \sum_{j=1}^{k} f(j|x_t^{(i)})\,\mathbb{E}[r_{t,j}^{(i)}]\right)$$

Further, using Lemma 2, we can write that, with probability $1 - \delta$ for all $f \in \mathcal{F}$ and all $\lambda \in \Delta(k)$, the following holds:

$$\widehat{\mathcal{L}}(\bar{f},\lambda) - \widehat{\mathcal{L}}(f,\bar{\lambda})$$

$$= \frac{1}{T}\sum_{t=1}^{T}\sum_{i=1}^{k}\lambda_i\left(\mathbb{E}[c_t^{(i)}] - \sum_{j=1}^{k} f_t(j|x_t^{(i)})\,\mathbb{E}[r_{t,j}^{(i)}]\right) - \frac{1}{T}\sum_{t=1}^{T}\sum_{i=1}^{k}\lambda_{t,i}\left(\mathbb{E}[c_t^{(i)}] - \sum_{j=1}^{k} f(j|x_t^{(i)})\,\mathbb{E}[r_{t,j}^{(i)}]\right)$$

$$\leq \frac{1}{T}\sum_{t=1}^{T}\sum_{i=1}^{k}\lambda_i\left(c_t^{(i)} - \sum_{j=1}^{k} f_t(j|x_t^{(i)})r_{t,j}^{(i)}\right) - \frac{1}{T}\sum_{t=1}^{T}\sum_{i=1}^{k}\lambda_{t,i}\left(c_t^{(i)} - \sum_{j=1}^{k} f(j|x_t^{(i)})r_{t,j}^{(i)}\right) + O\left(\sqrt{\frac{\log(k|\mathcal{F}|/\delta)}{T}}\right)$$

$$= \frac{1}{T}\sum_{t=1}^{T}\langle\lambda - \lambda_t, \ell_t'\rangle + \frac{1}{T}\sum_{t=1}^{T}\sum_{i=1}^{k}\langle\ell_t^{(i)}, f_t(x_t^{(i)}) - f(x_t^{(i)})\rangle + O\left(\sqrt{\frac{\log(k|\mathcal{F}|/\delta)}{T}}\right)$$

$$\leq \frac{\mathrm{Reg}_{\Lambda}(T)}{T} + \frac{\mathrm{Reg}_{\mathcal{F}}(T)}{T} + O\left(\sqrt{\frac{\log(k|\mathcal{F}|/\delta)}{T}}\right).$$

Since $\mathrm{Reg}_{\Lambda}(T) = O\left(\sqrt{T\log(k|\mathcal{F}|/\delta)}\right)$ with probability at least $1 - O(\delta)$, the result follows. □

**Lemma 4.** *Let* $V^\star = \inf_{f\in\mathcal{F}}\max_{\lambda\in\Delta_k}\mathcal{L}(f,\lambda)$ *be the optimal value of the saddle-point. Then Algorithm 1 converges to that value with probability at least* $1 - O(\delta)$*, that is,*

$$\max_{\lambda\in\Delta_k}\widehat{\mathcal{L}}(\bar{f},\lambda) \leq \widehat{V}^\star + \frac{\mathrm{Reg}_{\mathcal{F}}(T)}{T} + O\left(\sqrt{\frac{\log(k|\mathcal{F}|/\delta)}{T}}\right).$$

*This statement is true for either option A and option B.*

*Proof.* By Lemma 1, the following chain of inequalities holds

$$\inf_{f\in\mathcal{F}}\widehat{\mathcal{L}}(f,\bar{\lambda}) \leq \max_{\lambda\in\Delta_k}\inf_{f\in\mathcal{F}}\widehat{\mathcal{L}}(f,\lambda) = \widehat{V}^\star = \inf_{f\in\mathcal{F}}\max_{\lambda\in\Delta_k}\widehat{\mathcal{L}}(f,\lambda) \leq \max_{\lambda\in\Delta_k}\widehat{\mathcal{L}}(\bar{f},\lambda).$$

Rearranging terms yields

$$\widehat{\mathcal{L}}(\bar{f},\lambda) \leq \widehat{V}^\star + \max_{\lambda\in\Delta_k}\widehat{\mathcal{L}}(\bar{f},\lambda) - \inf_{f\in\mathcal{F}}\widehat{\mathcal{L}}(f,\bar{\lambda})$$

$$\leq \widehat{V}^\star + \frac{\mathrm{Reg}_{\mathcal{F}}(T)}{T} + O\left(\sqrt{\frac{\log(k|\mathcal{F}|/\delta)}{T}}\right). \qquad \text{(Lemma 3)}$$

□

## C.1 Analysis for Option A

**Lemma 5** (Concentration for option A). *For a fixed $\lambda$ and $f \in \mathcal{F}$, we have with probability at least $1 - \delta$*

$$\mathcal{L}_A(f, \lambda) - \widehat{\mathcal{L}}_A(f, \lambda) \leq O\left(\sqrt{\sum_{i=1}^k \frac{\lambda_i^2 \log(1/\delta)}{n_i}} + \max_i \frac{\lambda_i \log(1/\delta)}{n_i}\right)$$

$$\widehat{\mathcal{L}}_A(f, \lambda) - \mathcal{L}_A(f, \lambda) \leq O\left(\sqrt{\sum_{i=1}^k \frac{\lambda_i^2 \log(1/\delta)}{n_i}} + \max_i \frac{\lambda_i \log(1/\delta)}{n_i}\right) + \left|\sum_{i=1}^k \lambda_i \operatorname{bias}_A(i)\right|$$

*where*

$$\operatorname{bias}_A(i) = \frac{1}{n_i} \sum_{t=1}^{n_i} \left[\max_m v(\pi_m, x_t^{(i)}) - \mathbb{E}_{r_1, \ldots, r_m | x = x_t^{(i)}}\left[\max_m r_m\right]\right].$$

*Proof.* Consider and ordering of the samples in each augmented dataset and denote by $(x_t^{(i)}, r_{t,1}^{(i)}, \ldots, r_{t,m}^{(i)})$ the $t$-th sample in $\bar{D}_i$. Further define

$$\operatorname{bias}_A(i) = \frac{1}{n_i} \sum_{t=1}^{n_i} \left[\max_m v(\pi_m, x_t^{(i)}) - \mathbb{E}_{r_1, \ldots, r_m | x = x_t^{(i)}}\left[\max_m r_m\right]\right]$$

and

$$Y_{i,t} = \mathbb{E}_{r_1, \ldots, r_m | x = x_t^{(i)}}\left[\max_m r_m\right] - \max_m r_{t,m}^{(i)} - v(\pi_f, x_t^{(i)}) + \sum_{m=1}^k r_{t,m}^{(i)} f(m | x_t^{(i)})$$

Then we can decompose the difference in losses as

$$\mathcal{L}_A(f, \lambda) - \widehat{\mathcal{L}}_A(f, \lambda) = \sum_{i=1}^k \lambda_i \operatorname{bias}_A(i) + \sum_{i=1}^k \frac{\lambda_i}{n_i} \sum_{t=1}^{n_i} Y_{i,t}.$$

Since $Y_{i,t}$ are all independent from each other, we can bound the second term using concentration arguments as

$$\sum_{i=1}^k \frac{\lambda_i}{n_i} \sum_{t=1}^{n_i} Y_{i,t} \leq O\left(\sqrt{\sum_{i=1}^k \frac{\lambda_i^2 \log(1/\delta)}{n_i}} + \max_i \frac{\lambda_i \log(1/\delta)}{n_i}\right)$$

with probability at least $1 - O(\delta)$. Note that we can bound the negative, $-\sum_{t=1}^{n_i} Y_{i,t}$ analogously. Further, by Jensen's inequality, $\operatorname{bias}_A(i) \leq 0$ for all $i$. Combining these bounds yields the desired statement. $\square$

**Theorem 2** (Regret bound for Option A). *Assume that the function class $\mathcal{F}$ is convex. Then the solution $\bar{f}$ returned by Algorithm 1 with Option A satisfies with probability at least $1 - O(\delta)$*

$$\operatorname{reg}(\pi_{\bar{f}}, D_\lambda) \leq \operatorname{reg}(\pi_{pt}, D_\lambda) + \widehat{V}_A^\star + \frac{\operatorname{Reg}_{\mathcal{F}}(T)}{T} + O\left(\sqrt{\frac{\log(k|\mathcal{F}|/\delta)}{T}}\right) \tag{14}$$

$$+ O\left(\sqrt{\sum_{i=1}^k \frac{\lambda_i^2 \log(|\mathcal{F}|/\delta)}{n_i}} + \max_i \frac{\lambda_i \log(|\mathcal{F}|/\delta)}{n_i}\right) \tag{15}$$

*Further, if there exists an $f \in \mathcal{F}$ which perfectly predicts the maximum score per sample, i.e., $\sum_{i=1}^k \mathbb{E}_{(x, r_1, \ldots, r_k) \sim \bar{D}_i}[\max_m r_m] = \sum_{i=1}^k \mathbb{E}_{(x, r_1, \ldots, r_k) \sim \bar{D}_i} \mathbb{E}_{j \sim f(x)} r_j$, then $\widehat{V}_A^\star \leq 0$. If this only holds on a population level and for expected scores, i.e., $\sum_{i=1}^k \mathbb{E}_{x \sim D_i} \max_m v(\pi_m, x) = \sum_{i=1}^k \mathbb{E}_{x \sim D_i} v(\pi_f, x)$, then we can still bound $\widehat{V}_A^\star \leq \max_i |\operatorname{bias}_A(i)| + O\left(\frac{\log(|\mathcal{F}|/\delta)}{\sqrt{\min_i n_i}}\right)$.*

*Proof.* We can decompose the regret of $\bar{f}$ on $D_\lambda$ as

$$\text{reg}(\pi_{\bar{f}}, D_\lambda) = \max_{\pi \in \Pi} v(\pi, D_\lambda) - \mathbb{E}_{x \sim D_\lambda}\left[\max_m v(\pi_m, x)\right] + \mathbb{E}_{x \sim D_\lambda}\left[\max_m v(\pi_m, x)\right] - v(\pi_{\bar{f}}, D_\lambda)$$

$$= \max_{\pi \in \Pi} v(\pi, D_\lambda) - \mathbb{E}_{x \sim D_\lambda}\left[\max_m v(\pi_m, x)\right] + \mathcal{L}_A(\bar{f}, \lambda)$$

$$\leq \max_{\pi \in \Pi} v(\pi, D_\lambda) - \mathbb{E}_{x \sim D_\lambda}\left[\max_m v(\pi_m, x)\right] + \widehat{\mathcal{L}}_A(\bar{f}, \lambda)$$

$$+ O\left(\sqrt{\sum_{i=1}^k \frac{\lambda_i^2 \log(|\mathcal{F}|/\delta)}{n_i}} + \max_i \frac{\lambda_i \log(|\mathcal{F}|/\delta)}{n_i}\right) \qquad \text{(Lemma 5)}$$

To obtain a bound on $\widehat{\mathcal{L}}_A(\bar{f}, \lambda)$, we apply the game-theoretic arguments from Lemma 4

$$\widehat{\mathcal{L}}_A(\bar{f}, \lambda) \leq \widehat{V}_A^\star + \frac{\text{Reg}_{\mathcal{F}}(T)}{T} + O\left(\sqrt{\frac{\log(k|\mathcal{F}|/\delta)}{T}}\right)$$

and it only remains to control the optimal value of the game $\widehat{V}_A^\star$.

$$\widehat{V}_A^\star = \max_{\lambda \in \Delta_k} \inf_{f \in \mathcal{F}} \mathbb{E}_{(x, r_1, \ldots, r_k) \sim \bar{D}_\lambda} \mathbb{E}_{j \sim f(x)}\left[\max_m r_m - r_j\right]$$

$$\leq V_A^\star + \max_{\lambda \in \Delta_k}\left\{\left|\sum_{i=1}^k \lambda_i \text{ bias}_A(i)\right| + O\left(\sqrt{\sum_{i=1}^k \frac{\lambda_i^2 \log(|\mathcal{F}|/\delta)}{n_i}} + \max_i \frac{\lambda_i \log(|\mathcal{F}|/\delta)}{n_i}\right)\right\}$$

$$\leq V_A^\star + \max_i |\text{bias}_A(i)| + O\left(\frac{\log(|\mathcal{F}|/\delta)}{\sqrt{\min_i n_i}}\right)$$

$\square$

## C.2 Analysis for Option B

**Lemma 6** (Concentration for option B). *For a fixed $\lambda$ and $f \in \mathcal{F}$, we have with probability at least $1 - O(\delta)$*

$$\left|\mathcal{L}_B(f, \lambda) - \widehat{\mathcal{L}}_B(f, \lambda)\right| \leq O\left(\sqrt{\sum_{i=1}^k \frac{\lambda_i^2 \log(1/\delta)}{n_i}} + \max_i \frac{\lambda_i \log(1/\delta)}{n_i}\right)$$

*and*

$$\left|\mathcal{L}_B(f, \lambda) - \widehat{\mathcal{L}}_B(f, \lambda)\right| \leq O\left(\sum_{i=1}^k \lambda_i \sqrt{\frac{\log(1/\delta)}{n_i}}\right)$$

*Proof.* Consider a fixed $\lambda$, $f$ and $i \in [k]$. Order $\bar{D}_i$ arbitrarily and denote $(x_t, r_{t,1}, \ldots, r_{t,k})$ the $t$-th datapoint in $\bar{D}_i$. Then $Y_{i,t} = \mathbb{E}_{j \sim f(x_t)}[r_{t,i} - r_{t,j}]$ are i.i.d. random variables with mean $\mathbb{E}Y_{i,t} = v(\pi_i, D_i) - v(\pi_f, D_i)$. Since scores are bounded, $Y_{i,t}$ centered to its mean is sub-Gaussian and we can bound with probability at least $1 - \delta$

$$\mathcal{L}_B(f, \lambda) - \widehat{\mathcal{L}}_B(f, \lambda) = \sum_{i=1}^k \frac{\lambda_i}{n_i} \sum_{t=1}^{n_i} \left[\mathbb{E}Y_{i,t} - Y_{i,t}\right]$$

$$\leq O\left(\sqrt{\sum_{i=1}^k \sum_{t=1}^{n_i} \frac{\lambda_i^2}{n_i^2} \log(1/\delta)} + \max_i \frac{\lambda_i \log(1/\delta)}{n_i}\right)$$

$$= O\left(\sqrt{\sum_{i=1}^k \frac{\lambda_i^2 \log(1/\delta)}{n_i}} + \max_i \frac{\lambda_i \log(1/\delta)}{n_i}\right)$$

$\square$

**Lemma 7** (Value of the game for option B). *Let $V_B^\star = \inf_{f \in \mathcal{F}} \max_{\lambda \in \Delta_k} \mathcal{L}_B(f, \lambda)$ be the optimal value of the saddle-point. Assume that the function class $\mathcal{F}$ contains $f_\lambda$ for every $\lambda \in \Delta_k$, where $f_{\lambda, D}$ is defined as $f_\lambda(i|x) = \frac{\lambda_i D_i(x)}{\sum_{j=1}^k \lambda_j D_j(x)}$. Then the value of the game is non-positive, i.e., $V_B^\star \le 0$.*

*Proof.* Let $\lambda \in \Delta_k$ be arbitrary and consider $f(i|x) = \frac{\lambda_i D_i(x)}{\sum_{j=1}^k \lambda_j D_j(x)}$. We then have

$$
\begin{aligned}
\mathcal{L}_B(f, \lambda) &= v(\pi_{dom}, Q_\lambda) - v(\pi_f, D_\lambda) \\
&= \sum_{i=1}^k \lambda_i \sum_{x \in \mathcal{X}} D_i(x) \langle \pi_{dom}(x, i), r^\star(x) \rangle - \sum_{x \in \mathcal{X}} D_\lambda(x) \langle \pi_f(x), r^\star(x) \rangle \\
&= \sum_{i=1}^k \lambda_i \sum_{x \in \mathcal{X}} D_i(x) \langle \pi_{dom}(x, i), r^\star(x) \rangle - \sum_{x \in \mathcal{X}} \sum_{i=1}^k \lambda_i D_i(x) \langle \pi_i(x), r^\star(x) \rangle \\
&\quad\quad\quad\quad\quad\quad\quad\quad\quad\quad\quad\quad\quad\quad\quad\quad\quad\quad\quad\quad\quad\quad\quad\quad\quad\text{(definition of } f\text{)} \\
&= \sum_{i=1}^k \lambda_i \sum_{x \in \mathcal{X}} D_i(x) \langle \pi_{dom}(x, i) - \pi_i(x), r^\star(x) \rangle \\
&= 0 \quad\quad\quad\quad\quad\quad\quad\quad\quad\quad\quad\quad\quad\quad\quad\quad\quad\quad\quad\quad\quad (\pi_{dom}(x, i) = \pi_i(x))
\end{aligned}
$$

$\square$

**Theorem 3** (Regret bound for Option B). *Assume that the function class $\mathcal{F}$ is convex. Then the solution $\bar{f}$ returned by Algorithm 1 with Option B satisfies with probability at least $1 - O(\delta)$ for any fixed $\lambda$*

$$
\text{reg}(\pi_{\bar{f}}, D_\lambda) \le \sum_{i=1}^k \lambda_i \text{reg}(\pi_i, D_i) + \widehat{V}_B^\star + \frac{\text{Reg}_{\mathcal{F}}(T)}{T} + O\left(\sqrt{\frac{\log(k|\mathcal{F}|/\delta)}{T}}\right) \tag{16}
$$

$$
+ O\left(\sqrt{\sum_{i=1}^k \frac{\lambda_i}{n_i} \log \frac{|\mathcal{F}|}{\delta}} + \max_i \frac{\lambda_i}{n_i} \log \frac{|\mathcal{F}|}{\delta}\right) \tag{17}
$$

*Further, if the function class $\mathcal{F}$ contains $f_{\lambda, \widehat{D}}$ for every $\lambda \in \Delta_k$, where $f_{\lambda, \widehat{D}}$ is defined as $f_\lambda(i|x) = \frac{\lambda_i \widehat{D}_i(x)}{\sum_{j=1}^k \lambda_j \widehat{D}_j(x)}$, then $\widehat{V}_B^\star \le 0$. If this only holds on a population level, i.e., $\mathcal{F} \le \{f_{\lambda, D} : \lambda \in \Delta_k\}$, then we can still bound $\widehat{V}_B^\star = O\left(\sqrt{\frac{k \log(1/\delta)}{\min_i n_i}}\right)$.*

*Proof.* We can decompose the regret of $\bar{f}$ on $D_\lambda$ as

$$
\begin{aligned}
\text{reg}(\pi_{\bar{f}}, D_\lambda) &= \max_{\pi \in \Pi} v(\pi, D_\lambda) - v(\pi^\star, Q_\lambda) + v(\pi^\star, Q_\lambda) - v(\pi_{\bar{f}}, D_\lambda) \\
&= \max_{\pi \in \Pi} v(\pi, D_\lambda) - v(\pi^\star, Q_\lambda) + \mathcal{L}_B(\bar{f}, \lambda) \\
&\le \max_{\pi \in \Pi} v(\pi, D_\lambda) - v(\pi^\star, Q_\lambda) + \widehat{\mathcal{L}}_B(\bar{f}, \lambda) + O\left(\sqrt{\sum_{i=1}^k \frac{\lambda_i}{n_i} \log \frac{|\mathcal{F}|}{\delta}} + \max_i \frac{\lambda_i}{n_i} \log \frac{|\mathcal{F}|}{\delta}\right) \\
&\quad\quad\quad\quad\quad\quad\quad\quad\quad\quad\quad\quad\quad\quad\quad\quad\quad\quad\quad\quad\quad\quad\quad\quad\quad\quad\quad\quad\quad\text{(Lemma 6)}
\end{aligned}
$$

where the last inequality follows from a union bound over $f \in \mathcal{F}$ and holds with probability at least $1 - O(\delta)$. The first two terms can be upper-bounded by the regret of each expert policy $\pi_i$ on its own dataset, weighted by $\lambda$, i.e.,

$$
\max_{\pi \in \Pi} v(\pi, D_\lambda) - v(\pi^\star, Q_\lambda) = \max_{\pi \in \Pi} \sum_{i=1}^k \lambda_i \left(v(\pi, D_i) - v(\pi_i, D_i)\right) \le \sum_{i=1}^k \lambda_i \text{reg}(\pi_i, D_i).
$$

We now bound $\widehat{\mathcal{L}}_B(\bar{f}, \lambda)$ further by Lemma 4 with probability at least $1 - O(\delta)$ as

$$
\widehat{\mathcal{L}}_B(\bar{f}, \lambda) \le \widehat{V}_B^\star + \frac{\text{Reg}_{\mathcal{F}}(T)}{T} + O\left(\sqrt{\frac{\log(k|\mathcal{F}|/\delta)}{T}}\right).
$$

Plugging both bounds in the previous decomposition yields the desired bound. For the bound on $\bar{V}_B^\star$, we apply Lemma 7 on $\widehat{D}$ directly or on $D$ and apply Lemma 6 with a union bound over $\Delta_k$. $\square$

## C.3 Alternate Oracles

In this section we consider replacing the linear losses, $\ell_t^{(i)}$, from Algorithm 1 with a log-loss. Such a choice is natural whenever we consider $\mathcal{F}$ to be some family of Transformer networks for which modern ML packages use optimizers tailored to the cross-entropy loss. The losses constructed by Algorithm 1 are log-losses and so we need a different version of the Online Learning Oracle which we defined in Definition 2 The problem of Online Logistic Regression has been extensively studied in the online learning literature [Kakade and Ng, 2004; Xiao, 2009; McMahan and Streeter, 2012; Hazan et al., 2014; Foster et al., 2018; Shamir, 2020]. Using OLLO we can instantiate a new version of Algorithm 1 with the following losses for the min-player $\ell_t^{(i)'} = -\lambda_{t,i} e_{y_t^{(i)}}$ where $y_t^{(i)} \in \{j \in [k] : r_{t,j}^{(i)} = c_t^{(i)}\}$. Option A and Option B then correspond to the following two choices of $y_t^{(i)}$

$$y_t^{(i)} = \begin{cases} \operatorname{argmax}_{j \in [k]} r_{t,j}^{(i)} & \text{Option A} \\ r_{t,i}^{(i)} & \text{Option B.} \end{cases}$$

Next, we prove the counterpart to Lemma 4 for the classifier setting.

**Lemma 8.** *For any $\lambda \in \Delta(k)$ it holds that*

$$\sum_{t=1}^{T} \sum_{i=1}^{k} \lambda_i \left( c_t^{(i)} - \sum_{j=1}^{k} f_t(j|x_t^{(i)}) r_{t,j}^{(i)} \right)$$

$$\leq \min_{f \in \mathcal{F}} \sum_{t=1}^{T} \sum_{i=1}^{k} -\lambda_{t,i} \log(f(y_{t,i}^{(i)}|x_t^{(i)})) + \operatorname{Reg}_{\mathcal{F}}^{\mathsf{OLLO}}(T) + O(\sqrt{kT \log(k|\mathcal{F}|/\delta)}),$$

*with probability $1 - O(\delta)$.*

*Proof.* The definition of OLLO together with the standard analysis for the regret of the max-player imply the following holds with probability $1 - O(\delta)$

$$\sum_{t=1}^{T} \sum_{i=1}^{k} -\lambda_{t,i} \left( \log(f_t(y_t^{(i)}|x_t^{(i)})) - \log(f(y_t^{(i)}|x_t^{(i)})) \right) \leq \operatorname{Reg}_{\mathcal{F}}^{\mathsf{OLLO}}(T)$$

$$\sum_{t=1}^{T} \sum_{i=1}^{k} \lambda_i \left( c_t^{(i)} - \sum_{j=1}^{k} f_t(j|x_{i,t}) r_{t,j}^{(i)} \right)$$

$$- \sum_{t=1}^{T} \sum_{i=1}^{k} \lambda_{t,i} \left( c_t^{(i)} - \sum_{j=1}^{k} f_t(j|x_{i,t}) r_{t,j}^{(i)} \right) \leq O(\sqrt{kT \log(k|\mathcal{F}|/\delta)})$$

And so for any fixed $\lambda \in \Delta(k)$ we have

$$\sum_{t=1}^{T} \sum_{i=1}^{k} \lambda_i \left( c_t^{(i)} - \sum_{j=1}^{k} f_t(j|x_t^{(i)}) r_{t,j}^{(i)} \right)$$

$$\leq \sum_{t=1}^{T} \sum_{i=1}^{k} \lambda_{t,i} \left( c_t^{(i)} - \sum_{j=1}^{k} f_t(j|x_t^{(i)}) r_{t,j}^{(i)} \right) + O(\sqrt{kT \log(k|\mathcal{F}|/\delta)})$$

$$\leq \sum_{t=1}^{T} \sum_{i=1}^{k} \lambda_{t,i} r_{t,y_t^{(i)}}^{(i)} \left( 1 - f_t(y_t^{(i)}|x_t^{(i)}) \right) + O(\sqrt{kT \log(k|\mathcal{F}|/\delta)})$$

$$\leq \sum_{t=1}^{T} \sum_{i=1}^{k} -\lambda_{t,i} r_{t,y_t^{(i)}}^{(i)} \log(f_t(y_t^{(i)}|x_t^{(i)})) + O(\sqrt{kT \log(k|\mathcal{F}|/\delta)}).$$

for any $i$, where the last inequality uses $1 - x \leq -\log(x), x \in [0,1]$. The min-player regret guarantee together with the fact that $r_{t,y_t^{(i)}}^{(i)} \in [0,1]$ imply

$$\sum_{t=1}^{T} \sum_{i=1}^{k} \lambda_i \left( c_t^{(i)} - \sum_{j=1}^{k} f_t(j|x_t^{(i)}) r_{t,j}^{(i)} \right)$$

$$\leq \min_{f \in \mathcal{F}} \sum_{t=1}^{T} \sum_{i=1}^{k} -\lambda_{t,i} \log(f(y_{t,i}^{(i)}|x_t^{(i)})) + \operatorname{Reg}_{\mathcal{F}}^{\mathsf{OLLO}}(T) + O(\sqrt{kT \log(k|\mathcal{F}|/\delta)}).$$

$\square$

We need the following assumption to be able to guarantee concentration of the empirical log-loss to the expected log-loss.

**Assumption 1.** We assume that $\mathbb{E}_{i \sim \lambda, x^{(i)} \sim D_i, y^{(i)}}[-\log(f(y^{(i)}|x^{(i)}))^2] < +\infty$ for all $f \in \mathcal{F}, \lambda \in \Delta(k)$ and $y^{(i)}$ defined according to option A or option B.

This assumption is weaker than just assuming the log-loss is bounded which can be achieved by simply mixing $\varepsilon$ of the uniform distribution with the predictors in $\mathcal{F}$. Under Assumption 1 we have the following corollary which follows from Theorem 5 stated in Appendix E.

**Corollary 2.** *Suppose that Assumption 1 holds and fix an* $\varepsilon \in (0, 1)$. *Then w.p.* $1 - \delta$ *it holds that for all* $f \in \mathcal{F}$

$$\mathbb{E}_{i \sim \bar{\lambda}_i, x^{((i)} \sim \hat{D}_i} \left[ -\log(f(y^{(i)}|x^{(i)})) \right] - \mathbb{E}_{i \sim \bar{\lambda}_i, x^{((i)} \sim \bar{D}_i} \left[ -\log(f(y^{(i)}|x^{(i)})) \right]$$

$$\leq O \left( \log(T) \sqrt{\mathbb{E}_{i \sim \bar{\lambda}_i, x^{((i)} \sim \bar{D}_i} \left[ -\log(f(y^{(i)}|x^{(i)}))^2 \right] \frac{\log(\mathbb{E}[\mathcal{N}_{\infty}(-\log(\mathcal{F}), \varepsilon/2, x_{1:2T}^{(i)}, i \in [k])/\delta)]}{T}} \right) + \varepsilon$$

*where* $\mathcal{N}_{\infty}$ *denotes the* $\ell_{\infty}$ *covering number of the log-losses of* $\mathcal{F}$ *with respect to a sample from the empirical process induced by Algorithm 1 for* $2T$ *rounds.*

The main result for OLLO found in Theorem 4 below is very similar to that for OLO. The terms $\operatorname{Approx}(T), \operatorname{Conc}(\lambda, n)$ which depend on the oracle's regret and number of samples from the population still appear in the bound, however, $\operatorname{reg}(\pi_A^{\star}, D_\lambda) + \widehat{V}_A^{\star}$ is replaced by the log-loss of the best $f \in \mathcal{F}$ on the game: $\max_{i \in [k]} \mathbb{E}_{i \sim \bar{\lambda}, x^{(i)} \sim \bar{D}_i}[-\log(f(y^{(i)}|x^{(i)}))]$. This term depends on both the max-player's actions throughout the empirical game, $\bar{\lambda}$, and on the population through $\bar{D}_i, i \in [k]$. The only difference between Option A and Option B is that $y^{(i)} = \operatorname{argmax}_y r^{\star}(y, x^{(i)})$ for Option A and $y^{(i)} = i$ for Option B. Comparing Option A vs. Option B is similar to the comparison for the OLO. In particular $\min_{f \in \mathcal{F}} \mathbb{E}_{x^{(i)} \sim \bar{D}_i}[-\log(f(y^{(i)}|x^{(i)}))]$ is negligible under much stricter conditions for Option A, as outlined for OLOs, compared to Option B, where existence of an accurate domain classifier in $\mathcal{F}$ is sufficient. Finally, because the log-loss is potentially unbounded we need boundedness of its second moment for all $f \in \mathcal{F}$ to be able to show concentration of measure.

**Theorem 4.** *Fix* $\varepsilon \in (0, 1)$. *Under Assumption 1 with probability* $1 - \delta$ *it holds that for any* $\lambda \in \Delta(k)$ *and* $f \in \mathcal{F}$

$$1 \, \mathbb{E}_{i \sim \lambda, x \sim D_i}[c^{(i)} - \langle \bar{f}, r^{(i)} \rangle] \leq \mathbb{E}_{i \sim \bar{\lambda}, x^{(i)} \sim \bar{D}_i}[-\log(f(y^{(i)}|x^{(i)}))]$$

$$+ \frac{\operatorname{Reg}_{\mathcal{F}}^{\mathsf{OLLO}}(T)}{T} + O\left( \sqrt{\frac{\log(|\mathcal{F}|/\delta)}{T}} + \sqrt{\frac{\log(k/\delta)}{T}} \right.$$

$$+ \log(T) \sqrt{\mathbb{E}_{i \sim \bar{\lambda}_i, x^{((i)} \sim \bar{D}_i}[-\log(f(y^{(i)}|x^{(i)}))^2] \frac{\log(\Delta_T/\delta)}{T}} + \varepsilon \right)$$

$$+ O\left( \sqrt{\sum_{i=1}^{k} \frac{\lambda_i^2 \log(k|\mathcal{F}|T/\delta)}{n_i}} + \max_i \frac{\lambda_i \log(k|\mathcal{F}|T/\delta)}{n_i} \right),$$

*where* $\Delta_T$ *denotes the covering number from Corollary 2.*

*Proof.* We begin by arguing that

$$\mathbb{E}_{i \sim \lambda, x \sim \bar{D}_i}[c^{(i)} - \langle \bar{f}, r^{(i)} \rangle] \leq \mathbb{E}_{i \sim \bar{\lambda}, x^{(i)} \sim \bar{D}_i}[-\log(f(y^{(i)}|x^{(i)}))]$$

$$+ \frac{\operatorname{Reg}_{\mathcal{F}}^{\mathsf{OLLO}}(T)}{T} + O\left( \sqrt{\frac{\log(|\mathcal{F}|/\delta)}{T}} + \sqrt{\frac{\log(k/\delta)}{T}} \right)$$

$$+ O\left(\log(T)\sqrt{\mathbb{E}_{i\sim\bar{\lambda}_i,x^{((i)}\sim\bar{D}_i}\left[-\log(f(y^{(i)}|x^{(i)}))^2\right]\frac{\log(\Delta_T/\delta)}{T}+\varepsilon}\right)$$

This holds as follows. We combine the regret bound from Lemma 8 together with the concentration of Lemma 2 and Corollary 2.

Finally, we convert the LHS of the above lemma to a concentration over the population $\mathbb{E}_{i\sim\lambda,x\sim D_i}[c^{(i)}-\langle\bar{f},r^{(i)}\rangle]$ as follows. First note that for any fixed $f\in\mathcal{F}$:

$$\mathbb{E}_{i\sim\lambda,x\sim D_i}[c^{(i)}-\langle f,r^{(i)}\rangle]=\sum_{i=1}^{k}\frac{\lambda_i}{n_i}\sum_{j=1}^{n_i}c_j^{(i)}-\langle f,r_j^{(i)}\rangle.$$

We can then argue as in Lemma 2 that for all $\lambda\in\Delta(k)$ uniformly it holds that

$$\sum_{i=1}^{k}\frac{\lambda_i}{n_i}\sum_{j=1}^{n_i}\mathbb{E}[c_j^{(i)}-\langle f,r_j^{(i)}\rangle]-\sum_{i=1}^{k}\frac{\lambda_i}{n_i}\sum_{j=1}^{n_i}c_j^{(i)}-\langle f,r_j^{(i)}\rangle\leq O\left(\sqrt{\sum_{i=1}^{k}\frac{\lambda_i^2\log(k/\delta)}{n_i}}+\max_i\frac{\lambda_i\log(k/\delta)}{n_i}\right),$$

w.p. $1-O(\delta)$, where we use Bernstein's inequality instead of Hoeffding's inequality. An additional union bound over $\mathcal{F}$ now implies

$$\mathbb{P}\left(\sup_{\lambda\in\Delta(k),f\in\mathcal{F}}\sum_{i=1}^{k}\frac{\lambda_i}{n_i}\sum_{j=1}^{n_i}\mathbb{E}[c_j^{(i)}-\langle f,r_j^{(i)}\rangle]-\sum_{i=1}^{k}\frac{\lambda_i}{n_i}\sum_{j=1}^{n_i}c_j^{(i)}-\langle f,r_j^{(i)}\rangle\right.$$
$$\left.\geq\Omega\left(\sqrt{\sum_{i=1}^{k}\frac{\lambda_i^2\log(k|\mathcal{F}|/\delta)}{n_i}}+\max_i\frac{\lambda_i\log(k|\mathcal{F}|/\delta)}{n_i}\right)\right)\leq\delta.$$

Finally, we note that $\bar{f}\in\mathcal{F}$ by convexity of $\mathcal{F}$. and thus we need an extra union bound over $T$. This completes the proof of the theorem. $\qquad\square$

We can now show counterparts to Theorem 2 and Theorem 3.

**Corollary 3.** *Fix $\varepsilon\in(0,1)$. For any convex $\mathcal{F}$ for which Assumption 1 holds we have that for all $\lambda\in\Delta(k)$ with probability $1-\delta$*

$$\text{reg}(\pi_{\bar{f}},D_\lambda)\leq\min_{f\in\mathcal{F}}\mathbb{E}_{i\sim\bar{\lambda},x^{(i)}\sim\bar{D}_i}[-\log(f(y^{(i)}|x^{(i)}))]$$
$$+\frac{\text{Reg}_{\mathcal{F}}^{OLLO}(T)}{T}+O\left(\sqrt{\frac{\log(|\mathcal{F}|/\delta)}{T}}+\sqrt{\frac{\log(k/\delta)}{T}}\right)$$
$$+O\left(\log(T)\sqrt{\mathbb{E}_{i\sim\bar{\lambda}_i,x^{((i)}\sim\bar{D}_i}\left[-\log(f(y^{(i)}|x^{(i)}))^2\right]\frac{\log(\Delta_T/\delta)}{T}+\varepsilon}\right)$$
$$+O\left(\sqrt{\sum_{i=1}^{k}\frac{\lambda_i^2\log(k|\mathcal{F}|T/\delta)}{n_i}}+\max_i\frac{\lambda_i\log(k|\mathcal{F}|T/\delta)}{n_i}\right),$$

*where for Option A we have $y^{(i)}=\text{argmax}_{y\in[k]}r^\star(y,x^{(i)})$ and for Option B we have $y^{(i)}=i$.*

*Proof.* The definition of regret for Option A implies that

$$\text{reg}(\pi_{\bar{f}},D_\lambda)=v(\pi_A^\star,D_\lambda)-v(f,D_\lambda)=\mathbb{E}_{i\sim\lambda,x\sim D_i}[\max_{j\in[k]}v(\pi_j,x^{(i)})-v(\bar{f},x^{(i)})]$$
$$\leq\mathbb{E}_{i\sim\lambda,x\sim D_i}[\underset{y\in[k]}{\text{argmax}}\,r^\star(y,x^{(i)})-\langle\bar{f},r^\star(\cdot,x^{(i)})\rangle]$$
$$=\mathbb{E}_{i\sim\lambda,x\sim D_i}[c^{(i)}-\langle\bar{f},r^{(i)}\rangle].$$

The bound now follows from Theorem 4. For Option B we have a similar derivation with

$$\text{reg}(\pi_{\bar{f}},D_\lambda)=v(\pi_A^\star,D_\lambda)-v(\pi_{\bar{f}},D_\lambda)=\mathbb{E}_{i\sim\lambda,x\sim D_i}[v(\pi_i,x^{(i)})-v(\pi_{\bar{f}},x^{(i)})]$$

$$= \mathbb{E}_{i \sim \lambda, x \sim D_i, j \sim \pi_i(x^{(i)})} \left[ r^\star(j, x^{(i)}) - \sum_{l=1}^{k} \sum_{s=1}^{k} \bar{f}(s|x^{(i)}) \pi_s(l|x^{(i)}) r^\star(l, x^{(i)}) \right]$$

$$= \mathbb{E}_{i \sim \lambda, x \sim D_i, j \sim \pi_i(x^{(i)})} [c^{(i)} - \langle \bar{f}, r^{(i)} \rangle].$$

The bound again follows from Theorem 4. □

## D Experiments

### D.1 Experimental setup details

We identify a domain in the mix-instruct dataset by the prefix in the sample id, which is `dolly_15k`, `itwgpt4`, `laion`, `sharegpt`, or `unified_chip2`. The number of samples for each domain are 6858, 20754, 106, 2719 and 69563 respectively.

Results report the regret during training and Table 2 shows metrics aggregated from all checkpoints during training. Since we had metrics available for each checkpoint, there was no need for creating a single routing function, e.g., through model souping. The same evaluation procedure was used for all approaches including baselines.

All reported results are averaged over 5 replications.

We conducted our experiment on a cluster with 256 v5e TPUs. A single experiment run took about 2 hours. We estimate that all experiments involved in this research project took about 1 week of compute time.

### D.2 Additional results

In Figure 3, Figure 4 and Figure 5 we present the remaining comparisons for Algorithm 1 for OLO Option B and Option A, and for OLLO Option A respectively.

For Figure 3, OLO with Option B, we make the same observations as in the main text for OLLO with Option B, that Algorithm 1 tries to equalize the regrets across all domains which leads to reduced regret for Domain 4, but slightly higher regret for the remaining domains. The regret against the best per example expert is uniformly higher across domains compared to the baseline. This is not surprising as the objective which Algorithm 1 works with in this setting is not aimed at optimizing the regret against this stronger competitor.

For Figure 4, OLO with Option A, and Figure 5, OLLO with Option A we make the following observations. The regret against the best per example expert is decreased compared to the baseline across all domains for the OLO oracle and is decreased or remains the same for the OLLO oracle. This is in contrast with the results for Option B, where we see that the regret for domains increases when the domain weight is decreased by the max player. Further, we see that minimizing the regret to the best expert per token results in smaller regret against the domain experts as well. This is to be expected as the competitor defining the regret for Option A is stronger compared to that of Option B.

## E Unbounded loss bound

The following generalization bound follows directly Theorem 3 of [Cortes et al., 2021b]. It holds for any unbounded loss function with bounded second-moment. In particular, it can be applied to the $\log$ loss when the second-moment is bounded.

**Theorem 5.** *Fix $\varepsilon \in (0, 1)$. Then, for any hypothesis set $\mathcal{H}$ such that $\mathbb{E}_{x \sim \mathcal{D}}[\ell^2(h, x)] < +\infty$ for all $h \in \mathcal{H}$, the following holds with probably at least $1 - \delta$ over the draw of a sample of size $m$ from $\mathcal{D}$:*

$$\mathbb{E}_{x \sim \mathcal{D}}[\ell(h, x)] - \mathbb{E}_{x \sim S}[\ell(h, x)] \leq \gamma \sqrt{\mathbb{E}_{x \sim \mathcal{D}}[\ell^2(h, x)] \frac{\Delta_m}{m}} + \varepsilon,$$

*where $\Delta_m = \log \mathbb{E}[\mathcal{N}_\infty(\ell(\mathcal{H}), \frac{\varepsilon}{2}, x_1^{2m})] + \log \frac{1}{\delta}$, $\gamma = \Gamma_0 \left( \sqrt{\frac{\Delta_m}{m}} \right) = \mathcal{O}(\log m)$, and $\Gamma_0(\mu) = \frac{1}{2} + \sqrt{1 + \frac{1}{2} \log \frac{1}{\mu}}$ for any $\mu > 0$. $\mathcal{N}_\infty(\ell(\mathcal{H}), \frac{\varepsilon}{2}, x_1^{2m})$ represents the $\ell_\infty$-covering number of the $\ell$-losses associated with the hypotheses in $\mathcal{H}$ based on a sample of size $2m$, denote by $x_1^{2m}$, with a precision of $\frac{\varepsilon}{2}$.*

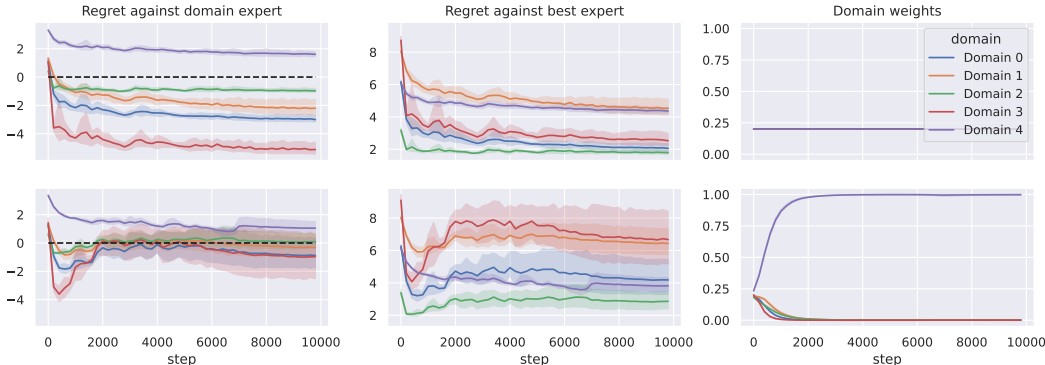

Figure 3: Comparison of Algorithm 1 with Option B (bottom) against baseline (top) without domain weight adjustment using linear loss, corresponding to an OLO oracle.

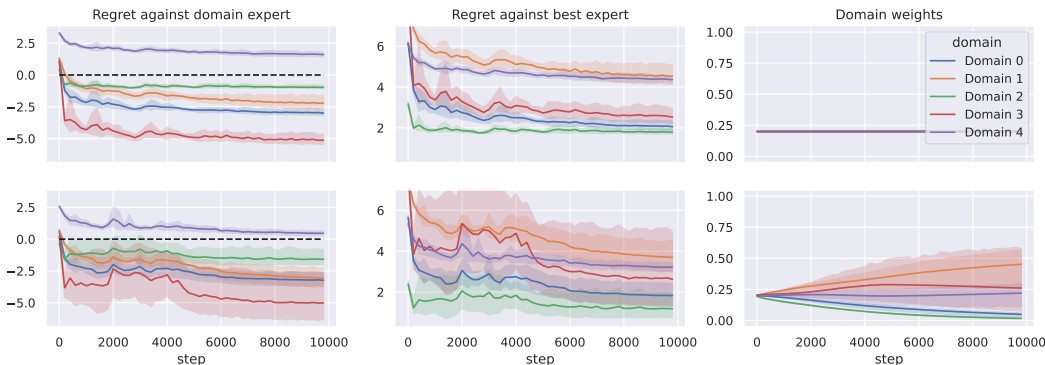

Figure 4: Comparison of Algorithm 1 with Option A (bottom) against baseline (top) without domain weight adjustment using linear loss, corresponding to an OLO oracle.

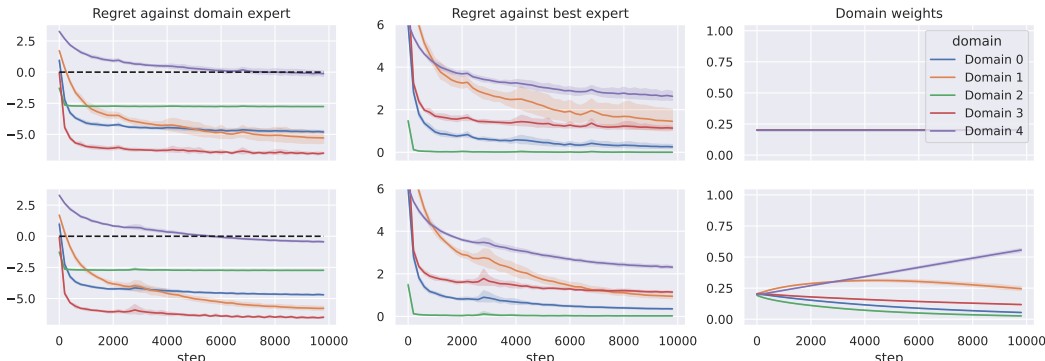

Figure 5: Comparison of Algorithm 1 with Option A (bottom) against baseline (top) without domain weight adjustment using log-loss, corresponding to an OLLO oracle.

In particular, we can choose $\varepsilon = \frac{1}{m}$ in the bound. The result generalizes to the case where only a higher-order moment of the loss (higher than 2) is bounded.

