# OpenReview forum: "Principled Model Routing for Unknown Mixtures of Source Domains"
_NeurIPS.cc/2025/Conference — NeurIPS 2025 poster_

### Official Review · Reviewer_83vD · 2025-06-21

**Clarity:** 3
**Significance:** 2
**Originality:** 3
**Rating:** 5
**Confidence:** 4

**Summary:**

This paper addresses the problem of routing inputs to the best-suited machine learning model from a set of domain-specialized models when the target distribution is an unknown mixture of source domains. It frames the problem as one of multiple-source domain adaptation (MSDA) and introduces a theoretically grounded algorithm to address it. Two algorithmic variants are proposed, and some empirical results are collected to demonstrate the approach.

**Questions:**

- Speaking of Option A, the authors claim that "this approach introduces additional bias when there is high variance in the expert scores for a given input" (lines 139 and 140). Can you explain the source/cause of this additional bias?
- On lines 171 and 172, the authors mention including a KL regularization term, to make the predicted distribution closer to "a uniform domain distribution or a given domain prior". Did you include this regularization in your experiments? Did you measure its impact on the results? Have you used any other prior besides the uniform?

**Ethical Concerns:**

["NO or VERY MINOR ethics concerns only"]

**Final Justification:**

As mentioned in my response to the Authors' rebuttal, they addressed one of my two major concerns favorably (lack of experiments with a single expert), while providing convincing arguments for not extending the experiments to more datasets. The answers to the remaining questions were also satisfactory. Therefore, I recommend acceptance of this manuscript.

**Limitations:**

The limitations are mentioned and acnowledged, although they are spread throughout the paper. I think the paper would benefit from having a dedicated section with a summary of those limitations.

**Paper Formatting Concerns:**

Nothing

**Quality:**

3

**Strengths And Weaknesses:**

Strengths:
- The problem is relevant and well-motivated, with clear distinctions from prior MoE and model routing works.
- The proposed method has a strong theoretical foundation, building on MSA theory to derive novel regret bounds that hold under reasonable assumptions. Although I did not do a step-by-step review for the proofs in the appendix, the arguments used in the proofs are correct and the obtained results are compatible with the existing prior work in DA, so I think it is safe to assume that they are at least mostly correct.
- The paper is technically high-quality, well written, well organized, and presents sufficient details and intuitions in most parts.

Weaknesses:
- The main weakness of the paper is its limited experimental scope. Although the authors claim to be mostly interested in making theoretical and algorithmic contributions, a more robust set of experiments could have been conducted.
- The evaluation should also be conducted in a practical usage scenario, e.g. where the learned router is used to select the best model for the given input and then a single prediction is generated from the chosen model and its quality is evaluated. I am willing to raise my score if such an experiment is added (or if the authors can convince me it would bring no additional value).
- The chosen reward metric (BLEU) has many known flaws and is no-longer the most reliable metric for virtually any NLP problem. It is therefore hard to understand its choice in this work.
- The distinction between OLO and OLLO oracles should be made more explicit, e.g. by providing the explicit equations for both.
- The choice of LLMs as the base models to which apply the proposed approach is a bit questionable. State of the art LLMs can handle several domains without much need for specialization / fine-tuning. This is not a major concern though, since the proposed approach is completely agnostic to the type of models it is being applied to.

---

> ### Author Rebuttal · Authors · 2025-07-31
>
> > The main weakness of the paper is its limited experimental scope. Although the authors claim to be mostly interested in making theoretical and algorithmic contributions, a more robust set of experiments could have been conducted.
>
> Thank you for your feedback. We appreciate the importance of broad empirical validation and agree that additional benchmarks would further strengthen the paper. That said, our primary goal in this work is to introduce a new perspective on model routing through a domain adaptation lens, along with a principled optimization framework supported by theoretical guarantees and a tractable implementation.
>
>
> To evaluate our approach empirically, we focused on the MixInstruct dataset, a recent and challenging benchmark specifically designed to study instruction tuning across diverse tasks. Our experiments demonstrate consistent gains over multiple baselines, including stronger alternatives to uniform routing, and we believe this provides solid evidence of the method’s practical effectiveness.
>
>
> We also note that identifying publicly available benchmark datasets that align well with our setting, involving input-dependent routing under unknown domain mixtures, is not straightforward. Existing benchmarks for multi-source adaptation or instruction tuning often lack the required structure for evaluating input-only routing or do not include domain/task metadata in a form compatible with our framework. We would be grateful for any suggestions the reviewer may have regarding additional suitable datasets.
>
>
> Extending our evaluation to other domains such as vision or multi-modal tasks is a promising direction that we intend to pursue. However, we believe the core contributions, a novel problem formulation, tractable algorithms with theoretical guarantees, and strong empirical results on a relevant benchmark, are already significant and valuable in their own right.
>
>
> >The evaluation should also be conducted in a practical usage scenario, e.g. where the learned router is used to select the best model for the given input and then a single prediction is generated from the chosen model and its quality is evaluated. I am willing to raise my score if such an experiment is added (or if the authors can convince me it would bring no additional value).
>
> At inference time we can certainly include an additional experiment where, instead of computing the prediction using the weighted mixture of experts, the router samples a single expert according to the learned distribution and outputs the prediction from that expert alone. This setup naturally aligns with the practical usage case the reviewer described.
>
>
> Importantly, this sampling-based strategy retains similar regret guarantees to those in Theorem 1 and Corollary 1. A standard martingale concentration argument shows that the cumulative reward obtained by sampling experts according to the learned distribution closely tracks the expected reward from using the mixture, ensuring the same asymptotic guarantees hold up to high-probability deviations.
>
>
> We will include this version of the algorithm, along with the corresponding theoretical justification, in the revised paper. We are also happy to include an experiment for the above version of the algorithm.
>
> We appreciate the reviewer’s openness to reconsidering their score and hope this addition addresses the concern.
>
> >The chosen reward metric (BLEU) has many known flaws and is no-longer the most reliable metric for virtually any NLP problem. It is therefore hard to understand its choice in this work.
>
> We agree that BLEU has well-known limitations. However, our choice was motivated by a desire to ensure comparability with MixInstruct (Jiang et al., 2023), which serves as the most relevant and recent benchmark for our setting and also uses BLEU as the reward metric. Grounding our experiments in this benchmark allowed us to align with prior work and highlight the benefits of our method under consistent evaluation conditions.
>
>
> That said, we agree that experimenting with more task-appropriate or learning-signal-aligned reward metrics is important and could further strengthen our results. We plan to explore alternative reward metrics in future versions and would be happy to consider specific suggestions from the reviewer.
>
> >The distinction between OLO and OLLO oracles should be made more explicit, e.g. by providing the explicit equations for both.
>
> We will provide explicit equations to further clarify Definitions 1 and 2, and will comment on the distinction.
>
> >The choice of LLMs as the base models to which apply the proposed approach is a bit questionable. State of the art LLMs can handle several domains without much need for specialization / fine-tuning. This is not a major concern though, since the proposed approach is completely agnostic to the type of models it is being applied to.
>
> This really depends on the domains, for extended discussion please see our response to Reviewer y9hx.
>
>
> >Speaking of Option A, the authors claim that "this approach introduces additional bias when there is high variance in the expert scores for a given input" (lines 139 and 140). Can you explain the source/cause of this additional bias?
>
> In essence, bias_A quantifies the bias from estimating the maximum expected score by taking the maximum over realizations, thus exchanging the order of the expectation and maximum in the expression. This term is favorably small, e.g. when all scores are deterministic or they are random but there is always a single fixed model that is best. See also our discussion regarding Option A in the paragraph between lines 259-278.
>
>
> >On lines 171 and 172, the authors mention including a KL regularization term, to make the predicted distribution closer to "a uniform domain distribution or a given domain prior". Did you include this regularization in your experiments? Did you measure its impact on the results? Have you used any other prior besides the uniform?
>
> Yes, we included the KL regularization term in our experiments, using a uniform domain distribution as the prior and a regularization weight of 1.7. This choice was motivated by preliminary experiments where omitting the KL term led to the following behavior: after several thousand training steps, we observed that the instantaneous regret began to increase, indicating that the router was overfitting or drifting toward degenerate policies.
>
>
> The KL regularization serves to stabilize training by encouraging the learned routing distribution to remain close to a uniform baseline, effectively shrinking the policy class and preventing overconfident or brittle behavior in early stages of training. While we did not carry out an extensive "ablation" across different priors, we did experiment briefly with removing the regularization altogether and found it to negatively affect stability.
>
>
> We have not explored alternative priors beyond the uniform in this work, but doing so, e.g., by incorporating domain frequency estimates or prior knowledge, could be a promising direction for future research. We will clarify this in the revised version and include a brief discussion of the KL term’s impact.

---

> > ### Author Response · Authors · 2025-08-02
> >
> > We wanted to quickly update Reviewer 83vD on the proposed efficient algorithm which samples an expert from the learned distribution over domains. We were able to run an extra set of experiments for the setting of using log-loss with Option A. The regret curves look very similar to that of the non-sampling algorithm presented in the paper. We also add the corresponding results to that of Table 2:
> >
> > Loss for f--|-Option---------|-------regret vs best expert--|--regret vs domain expert
> >
> > log-----------|  A (sampling) |-------1.974851-------------------|---------- -0.784192
> >
> > Somewhat surprisingly we observe better performance compared to the non-sampling version presented in the paper.

---

> > ### Comment · Reviewer_83vD · 2025-08-04
> > **Answer to Author's rebuttal**
> >
> > Dear Authors,
> >
> > Thank you for your detailed response to my questions. Although I would have liked to see in your rebuttal some results of the single expert experiment, I appreciate your detailed explanation and the clear argument you provide to extend the derived theoretical guarantees to this scenario. Therefore, as mentioned in my initial review, I will raise my score to an "Accept".

---

### Official Review · Reviewer_APhc · 2025-06-27

**Clarity:** 4
**Significance:** 3
**Originality:** 3
**Rating:** 4
**Confidence:** 2

**Summary:**

The paper addresses the problem of routing inputs to the best model from a pool of domain experts when the test-time distribution is an unknown mixture of the source domains. The authors formulate the problem as a multiple-source domain adaptation (MSA) problem and propose an algorithm based on minimax-regret optimization.

They provide theoretical guarantees and validate the approach empirically.

**Questions:**

I refrain from posing questions due to my low confidence, and apologize for this to the AC and the other reviewers.

**Ethical Concerns:**

["NO or VERY MINOR ethics concerns only"]

**Final Justification:**

My initial review noted my low confidence in assessing novelty of the paper, as the specific subfield is outside my area of expertise.

The assessments from other reviewers have partially resolved this uncertainty. After reading their comments I have raised my confidence score while maintaining my recommendation for acceptance.

**Limitations:**

Yes

**Quality:**

3

**Strengths And Weaknesses:**

## Strengths
- Presentation: The paper is exceptionally well-written, clear, and logically structured.
- Theoretically Grounded: The work is built on a solid theoretical foundation. The minimax-regret formulation seem sound

I appreciate the note on applying the framework to discriminative models.

## Weaknesses
- None of the paper as far as I am aware. My primary limitation as a reviewer, detailed below, is in assessing the novelty and significance

I have spent considerable time on this paper. Its quality is apparent. The formalism, the derivations, and the theoretical analysis all appear to be correct. However, the specific field of multiple-source domain adaptation is outside my expertise. While I can attest to quality of the writing, I cannot comment on the novelty or significance of the work.

This appears to be a high-quality paper, and I recommend accepting it. My confidence is, however, low.

---

> ### Author Rebuttal · Authors · 2025-07-31
>
> Response: Thank you very much for your thoughtful review and for taking the time to carefully read through our paper. We greatly appreciate your positive assessment of the writing, formalism, and theoretical analysis.
> We understand that the specific topic of multi-source domain adaptation (MSA) may fall outside your primary area of expertise. To help clarify the novelty and significance of our contribution within this field, we would like to briefly summarize the key aspects:
>
>
>  - Our work introduces a new input-dependent framework grounded in a formal min-max formulation, which differs from standard MSA settings that typically assume access to domain labels or covariate shift assumptions.
>
>
>  - We present tractable algorithms based on online learning oracles, enabling efficient implementation in large-scale settings, a practical gap in much of the prior MSA literature.
>
>
>  - Our theoretical guarantees are derived under general conditions and extend existing results by removing some assumptions typically made in prior work (e.g., domain or task supervision).
>
>
>  - Empirically, we evaluate our method on a benchmark task introduced in recent state-of-the-art work (Jiang et al., 2023) and compare against multiple stronger baselines beyond uniform routing.
>
> We hope this helps situate our contribution within the broader MSA literature and reassures you of its relevance. Once again, we are grateful for your recommendation and the time you dedicated to this review.

---

> > ### Comment · Reviewer_APhc · 2025-08-04
> >
> > I thank the authors for their rebuttal.
> >
> > After reading the other reviews, I became more confident in my initial assessment.
> >
> > I retain my recommendation for acceptance.

---

### Official Review · Reviewer_BcKM · 2025-07-05

**Clarity:** 3
**Significance:** 3
**Originality:** 3
**Rating:** 4
**Confidence:** 2

**Summary:**

This submission investigates the model routing problem for testing data that are unknown mixtures of source data domains. The authors formulate this challenge as a multi-source domain adaptation problem and propose a novel and scalable routing algorithm that can achieve regret bounds tightly linked to the online learning oracles. Strong theoretical guarantees ensure the robustness of the proposed model routing strategy. Empirical results on LLM instruction tasks also demonstrate its advantage over the baseline of uniform routing.

**Questions:**

Check the weaknesses.

**Ethical Concerns:**

["NO or VERY MINOR ethics concerns only"]

**Final Justification:**

I would like to maintain my original rating.

**Limitations:**

Yes.

**Paper Formatting Concerns:**

No concerns.

**Quality:**

3

**Strengths And Weaknesses:**

## Strengths

1. The studied model routing or selection problem is of great significance in improving model performance in complex real-world deployment scenarios.

2. Both the perspective of taking model routing as multi-source domain adaptation and the proposed routing algorithm are novel.

3. The presentation is clear, and the theoretical analysis is very solid.

## Weaknesses

1. Although I acknowledge that the primary contributions of the paper are theoretical and algorithmic, current empirical results are not extensible. First, it only demonstrates the advantage of the proposed routing algorithm over the vanilla uniform routing strategy. It would enhance the empirical justification by providing results of advanced model selection or routing baselines. Second, since multi-source domain adaptation is widely studied in vision tasks [Hoffman et al., 2021],  adding empirical results of vision tasks would also enhance the justification.

2. The efficiency analysis of the algorithm is not discussed. Whether efficiency may be a concern depends on the problem setting. What is the sample size of the target domain? Is target access and evaluation offline or online? What if the target domain is not a trivial mixture of source domains, i.e., the target domain consists of various out-of-distribution data?

---

> ### Author Rebuttal · Authors · 2025-07-31
>
> >First, it only demonstrates the advantage of the proposed routing algorithm over the vanilla uniform routing strategy. It would enhance the empirical justification by providing results of advanced model selection or routing baselines.
>
> Thank you for the suggestion. We would like to clarify that our method is not only compared against uniform routing, but also against two stronger baselines for router learning without domain adaptation: direct accuracy maximization and cross-entropy minimization. Across all settings, we observe consistent performance gains, demonstrating the practical advantage of our approach beyond naive or uniform strategies.
>
>
> While we agree that additional experiments are always valuable, scaling them, especially in large-scale settings, is computationally expensive. To ensure fair and meaningful comparisons, we designed our evaluation using the benchmark task introduced in Jiang et al. (2023), which reflects current state-of-the-art experimental setups for this problem. We believe these experiments are sufficient to validate the effectiveness of our method in a realistic and widely accepted setting.
>
>
> That said, we appreciate the suggestion to include more advanced model selection or routing baselines. We consider this a valuable direction for follow-up work, particularly as new methods emerge. Additionally, incorporating multi-source domain adaptation tasks from vision, as studied in works such as Hoffman et al. (2021), could further enrich our empirical validation. We plan to explore such extensions in future iterations of this work.
>
>
> >The efficiency analysis of the algorithm is not discussed. Whether efficiency may be a concern depends on the problem setting.
>
> Efficiency is not a concern in our experimental setup even though we use a relatively large LLM for the policy class. This is because the OLO and OLLO have particularly nice forms where we can use the simple closed form update of Hedge for the max player and use standard first order optimization methods for the min-player. Overall the computational complexity is no higher than mini-batched versions of gradient descent for optimizing LLMs.
>
> >What is the sample size of the target domain?
>
> The number of samples for each domain are 6858, 20754, 106, 2719 and 69563 respectively. These details are already included in Appendix D.1
>
>
> >Is target access and evaluation offline or online?
>
> The evaluation is online, that is we plot the current instantaneous regret.
>
>
> >What if the target domain is not a trivial mixture of source domains, i.e., the target domain consists of various out-of-distribution data?
>
> While a full analysis of this setting is beyond the scope of our work, we note that, similar to prior results in the MSA literature (Mansour et al., 2021; Hoffman et al., 2021; Cortes et al., 2021b), one can show that if the R'enyi divergence between the target distribution and the convex hull of the source domain distributions is sufficiently small, then our performance guarantees remain approximately valid and practically meaningful.

---

> > ### Comment · Reviewer_BcKM · 2025-08-05
> >
> > Thank you for the clarification. Most of my initial questions have been addressed. I would like to maintain my original rating.

---

### Official Review · Reviewer_y9hx · 2025-07-07

**Clarity:** 3
**Significance:** 1
**Originality:** 1
**Rating:** 3
**Confidence:** 3

**Summary:**

The paper introduces an algorithm for predictive model routing in the presence of unknown mixtures of source domains, framed as a multiple-source domain adaptation problem. The authors leverage minimax regret optimization and online learning oracles to design a routing function that selects the best model for a given input among several domain-specialized models. The approach is supported by theoretical regret bounds  and evaluated empirically on the MixInstruct benchmark using LLMs.

**Questions:**

I suggest analyzing routing failures in existing LLMs to motivate scenarios where predictive, input-only routing with unknown mixtures is practically impactful.

**Ethical Concerns:**

["NO or VERY MINOR ethics concerns only"]

**Limitations:**

yes

**Quality:**

3

**Strengths And Weaknesses:**

strengths:
- The paper provides detailed, formal regret bounds for both online learning oracles and online Log-Loss oracle -based algorithms grounded in established multiple-source domain adaptation literature.
- The paper openly discusses limitations of assumptions

weakness:
- LLMs typically have prior knowledge of application tasks and their relative frequencies (contrary to the paper's assumption on unknown domain mixtures
- The proposed framework targets robustness to arbitrary mixtures. In practice, adversarial or drastically shifting mixtures are rare, and the central technical contributions seem tailored to an artificially difficult setting rather than one current generative ML systems face.
- The saddle-point approach and dueling no-regret learners follow closely from the cited literature, the problem framing and regret decomposition rely directly on classic MSA theory. The paper primarily applies these known tools to this contrived setting without introducing theoretical innovations beyond careful integration.

---

> ### Author Rebuttal · Authors · 2025-07-31
>
> >LLMs typically have prior knowledge of application tasks and their relative frequencies (contrary to the paper's assumption on unknown domain mixtures
>
> >The proposed framework targets robustness to arbitrary mixtures. In practice, adversarial or drastically shifting mixtures are rare, and the central technical contributions seem tailored to an artificially difficult setting rather than one current generative ML systems face.
>
> We respectfully disagree with the claim that the assumption of an unknown domain mixture is invalidated by the use of large language models (LLMs). While LLMs are indeed pretrained on broad web-scale corpora (e.g., Wikipedia, StackOverflow, news, books, code) the specific domain weights used in training are carefully handcrafted and training is very sensitive to the mixture choices. For example improving pre-trained performance on coding can adversely affect the performance on natural language tasks such as text summarization or reasoning on math datasets. Furthermore, pretraining on a wide range of domain mixtures does not eliminate the need to explicitly model uncertainty over domain mixtures at deployment time.
>
> In fact, the unknown mixture assumption remains critical in many practical settings:
>
>
> - Specialized deployment domains: LLMs are increasingly deployed in high-stakes areas such as medical triage, legal document analysis, or scientific assistant tools. These domains often involve highly skewed, out-of-distribution inputs. For instance, a model used in an emergency room chatbot will encounter a mixture of rare and critical conditions not seen in pretraining, with dramatically different costs of error.
>
>
> - Dynamic task distributions: Applications like customer support, AI tutoring systems, or internal enterprise tools often face time-varying or user-dependent mixtures of sub-tasks (e.g., billing issues vs. technical queries). These subpopulations shift over time or across users and are not captured by static pretraining priors.
>
>
> - Multi-agent or expert systems: In systems involving deferral to humans, collaboration between models, or fallback mechanisms, modeling the domain/task distribution is essential. For example, if an AI assistant can defer to a human or another model, it must reason about the uncertainty of task types and associated costs—something naturally handled via mixture models.
>
>
> - Fairness and group robustness: In fairness-sensitive applications (e.g., lending or hiring), subpopulations (e.g., based on age, location, or socioeconomic status) may have distinct conditional distributions and label noise characteristics. The test-time population may be an unknown mixture of these, and performance guarantees require reasoning under distributional uncertainty.
>
>
> - Security and robustness: Adversarial attacks or distributional shifts (e.g., bots or prompt hacking) can skew the input distribution in unpredictable ways. Modeling the deployment distribution as an unknown mixture helps reason about worst-case behavior and supports robust optimization.
>
>
> These and other examples demonstrate that, even though LLMs encode useful inductive biases, they do not explicitly model the task or domain distribution at inference time, nor do they adapt to new mixtures unless fine-tuned or combined with adaptive mechanisms such as routing, calibration, or deferral.
>
>
> Moreover, even during fine-tuning, similar mixture-related challenges arise: practitioners must balance multiple competing objectives, e.g., truthfulness, creativity, helpfulness, safety (see, e.g., [1–4]). These can be viewed as distinct domains, with specialized models or loss functions acting as domain experts. Modeling such cases as mixtures is not only natural but essential for scalable adaptation and control.
>
>
> Our framework explicitly captures this setting, providing a principled basis for learning and decision-making under unknown and heterogeneous domain mixtures.
>
> >The saddle-point approach and dueling no-regret learners follow closely from the cited literature, the problem framing and regret decomposition rely directly on classic MSA theory. The paper primarily applies these known tools to this contrived setting without introducing theoretical innovations beyond careful integration.
>
> We appreciate the reviewer’s feedback and agree that the online min-max framework we use draws on known tools from the literature, including dueling no-regret learners and classic MSA theory. We do not claim to introduce fundamentally new techniques for analyzing such games, and we explicitly acknowledge the standard nature of the regret decomposition in the paragraph starting on line 151.
>
>
> That said, we would like to emphasize that the key novelty of our work lies in the formulation and tractable instantiation of the min-max optimization problem in Equation (2), which is distinct from classical MSA settings. While our setup is inspired by the MSA framework, the problem we introduce is not a direct or trivial extension. Specifically:
>
>
> The problem structure in Equation (2) is tailored to a setting involving instance-dependent deferral decisions, requiring a careful rethinking of both the minimization and maximization objectives.
> We propose two computationally tractable versions, Option A and Option B, that admit practical implementations using first-order online learning oracles. These variants are significantly simpler to implement compared to typical MSA approaches, which often require solving non-convex DC (difference-of-convex) programs that are computationally more intensive and less scalable.
>
>
> Thus, our contribution is not in reinventing the theory behind online min-max games, but in developing a new problem formulation and practical algorithmic strategy that enables efficient training in settings where MSA would be theoretically applicable but operationally cumbersome. We believe this design perspective is both novel and valuable, especially in domains where computational simplicity is essential. Additionally, we provide a series of theoretical guarantees supporting our suggested solutions.
>
> >I suggest analyzing routing failures in existing LLMs to motivate scenarios where predictive, input-only routing with unknown mixtures is practically impactful.
>
> Thank you for the suggestion. We agree that analyzing failure modes in LLMs can help motivate the need for predictive, input-only routing under unknown mixtures.
>
>
> There is growing empirical evidence that LLMs struggle to specialize simultaneously across diverse domains and objectives. For example, several recent works [1–5] highlight issues such as catastrophic forgetting during continual fine-tuning [5], instability in multi-objective alignment [1–4], and interference between domain-specific capabilities. These failures become particularly pronounced when LLMs are expected to handle heterogeneous tasks without explicit task supervision or routing.
>
>
> These observations underscore the need for modular systems with effective input-level routing mechanisms—especially in settings where the input distribution is a latent mixture over tasks or domains. In such settings, predictive routing (based solely on input features) offers a practical solution that avoids costly retraining or fine-tuning, and our framework provides a principled way to learn such routing policies under domain uncertainty.
>
>
> We will revise the text to better highlight this motivation and thank the reviewer again for pointing out this connection.
>
>
> [1] Wang, Kaiwen, et al. "Conditional language policy: A general framework for steerable multi-objective finetuning." arXiv preprint arXiv:2407.15762 (2024).
>
> [2] Guo, Yiju, et al. "Controllable preference optimization: Toward controllable multi-objective alignment." arXiv preprint arXiv:2402.19085 (2024).
>
> [3] Jang, Joel, et al. "Personalized soups: Personalized large language model alignment via post-hoc parameter merging." arXiv preprint arXiv:2310.11564 (2023).
>
> [4] Rame, Alexandre, et al. "Rewarded soups: towards pareto-optimal alignment by interpolating weights fine-tuned on diverse rewards." Advances in Neural Information Processing Systems 36 (2023): 71095-71134.
>
> [5] Luo, Yun, et al. "An empirical study of catastrophic forgetting in large language models during continual fine-tuning." arXiv preprint arXiv:2308.08747 (2023).

---

> > ### Comment · Reviewer_y9hx · 2025-08-08
> >
> > The rebuttal makes reference to practical settings where mixture assumption remains critical. Yet we know in practice LLMs are being used in all those settings without the need to "model uncertainty over domain mixtures at deployment time.".
> >
> > Algorithmically, the building blocks of their methods are all directly from existing MSA + minimax regret literature. Equation (2) is just the standard minimize worst-case regret over domain mixtures formulation, already found in Cortes et al. (2021b) and Mohri et al. (2019). The "instance-dependent deferral" here is simply routing to experts - which in the MSA literature corresponds to input-dependent mixture weights.
> >
> > Option A and B are not fundamentally new relaxations; they are standard comparator substitutions to simplify regret computation. This is common in practical MSA implementations to avoid inner optimizations.
> > The claim that standard MSA methods "often require solving non-convex DC programs" is misleading; many MSA algorithms (esp. online oracles + Hedge) are already convex and scalable, just like their approach.

---

> > > ### Author Response · Authors · 2025-08-08
> > >
> > > We thank the reviewer for their comments and the opportunity to clarify our contributions. Below, we address each point in turn.
> > >
> > > > Yet we know in practice LLMs are being used in all those settings without the need to 'model uncertainty over domain mixtures at deployment time.
> > >
> > > We agree that, in practice, LLMs are indeed used for prompts drawn from mixtures of distributions. However, we respectfully disagree with the assertion that there is no need to explicitly model uncertainty in such cases. When prompts originate from combinations within the convex hull of the training distributions, performance can degrade if relying solely on single-domain models. For example, prompts combining both coding and math elements are common, and responses may be suboptimal when handled by only a coding model or only a math model. Our formulation directly addresses such scenarios.
> > >
> > > > Equation (2) is just the standard minimize worst-case regret over domain mixtures formulation, already found in Cortes et al. (2021b) and Mohri et al. (2019).
> > >
> > > Equation (2) is not the same as the formulations in Cortes et al. (2021b) or Mohri et al. (2019). In those works, the min–max problem is formulated in terms of the loss, not the regret. This distinction is significant: our consideration of regret, rather than loss, is precisely what motivates the need for Options A and B in our approach.
> > >
> > > > Option A and B are not fundamentally new relaxations; they are standard comparator substitutions to simplify regret computation. This is common in practical MSA implementations to avoid inner optimizations.
> > >
> > > To our knowledge, this characterization is not accurate. Prior MSA formulations address the loss, not the regret. As noted above, it is our regret-based formulation that specifically necessitates these options.
> > >
> > > > The claim that standard MSA methods 'often require solving non-convex DC programs' is misleading; many MSA algorithms (esp. online oracles + Hedge) are already convex and scalable, just like their approach.
> > >
> > > Our statement referred specifically to the MSA algorithms in Cortes et al. (2021b) and Hoffmann et al. (2021). The work of Mohri et al. (2019) indeed provides convex and scalable algorithms (e.g., Mirror Descent) for a different setting, again, one based on a min–max loss formulation, not regret.

---

### Comment · Area_Chair_3vfg · 2025-08-03
**author-reviewer discussion**

Dear reviewers,

The rebuttal provided by the authors is now available.
It is now time to read authors' answers and check if your questions/issues were addressed and possibly engage discussion to ask for more  information as soon as possible.
I also encourage you to read the other reviews to see the point of view of the other reviewers.


Many thanks for your participation.

Best regards,
AC.

---

### Decision · Program_Chairs · 2025-09-17

**Decision:**

Accept (poster)

**Comment:**

This paper focuses on the multiple-source domain adaptation problem when target data follow an unknown mixture of source data domains. The authors propose an algorithm based on minimax regret optimization and online learning oracles that selects for each input  the best model among several trained (source) models, 2 variants (options) are proposed to improve the practical efficiency. Another contribution is the theoretical analysis in the form of regret bound(s) on the performance on new domains is proposed, expressed as a convex combination of the best regrets in the source domains accompanied with a concentration term that diminishes when the concentration of source data increases.  An empirical evaluation studying the behavior of the method is also proposed.

Based on the released reviews, the following strengths have been identified:
-Paper well written, presence of a discussion on the limitations.
-Strong theoretical foundations.
-The proposed selection model is innovative and interesting.
-The model is interesting for addressing complex scenarios.

The following weaknesses have also been identified:
-The proposed setting is limited to arbitrary mixtures.
-Limited theoretical innovation.
-Limited empirical evaluation.
-Efficiency not sufficiently discussed.

During rebuttal, authors provided multiple answers to reviewers’ issues, many of them were addressed but some remarks related to limited experimental evaluation and limited setting were maybe not completely addressed. During discussion and final evaluation, the reviewers made the following feedbacks.


-Reviewer APhc reminded that for him the paper is well-written and the theory presented is sound. While the empirical evaluation is limited and efficiency is not discussed, he did not consider these to be grounds for rejection for a theoretical paper. Concerning the assessment of novelty, he was initially not confident as the specific subfield is outside his area of expertise, but he mentioned that the assessments from other reviewers have partially resolved this uncertainty and after reading their comments he was more confident on his evaluation. Notably, he carefully considered Reviewer y9hx comments and while the presented theory is an extension of existing work he still leaned towards acceptance. Eventually, he maintained his recommendation as borderline accept.
-Reviewer 83vD mentioned that authors’ rebuttal addressed one of his two major concerns favorably (lack of experiments with a single expert), while providing convincing arguments for not extending the experiments to more datasets. He also indicated that the answers to the remaining questions were also satisfactory. During discussion, he mentioned that he understood the concerns of Reviewer y9hx that the theoretical contribution of this paper might be somewhat incremental and the underlying assumptions might be too pessimistic in many cases, but he was still convinced that the theoretical results represent an interesting contribution to the community. Then he proposed the direct acceptance of this paper and supported it.
-Reviewer BcKM indicated that most of his initial questions have been addressed. He would like to maintain his original rating of borderline acceptance but did not support the paper beyond.
-Reviewer y9hx argued in his review that the building of the methods come directly from existing Multiple-source Domain Adaptation and minimax regret literature. Eq.(2) for example is a standard minimize worst-case regret over domain mixtures formulation adapted notably from Cortes et al. (2021b) and that the 2 options proposed are common to simplify regret computation. However, the reviewer did not participate actively to discussion. Note tha  I skipped here the remarks on LLMs which I found not directly relevant for the evaluation of the paper.

Authors have answered to reviewer y9hx that they did not aim at fundamentally introducing new techniques, but a formulation and a tractable instantiation of the min-max problem of Eq.(2). A novelty of the work is the formulation in terms of regret rather than loss (in comparison to the literature). One important strength of the paper lie the theoretical results which has been recognized as a strength by the reviewer. The other reviewer have a positive evaluation of the paper and at least 2 of them (Reviewer 83vD and APhc) felt that the strengths of the paper outweighed the weaknesses mentioned by reviewer y9hx. Reviewer BcKM maintained his borderline accept evaluation while reviewer y9hx did not react against acceptance. I read the paper and I saw that the theoretical part was a strong point of the contribution.

I then decided to follow the line of the majority of the reviewers by proposing acceptance.
I recommend to the authors to integrate in their revised version the elements provided notably to reviewer 83vD and also those clarifying the novelty of the contribution that appear along the discussions with reviewers.